# A homologue of the Parkinson's disease-associated protein LRRK2 undergoes a monomer-dimer transition during GTP turnover

Egon Deyaert[1,2], Lina Wauters[1,2,3], Giambattista Guaitoli[4,5], Albert Konijnenberg[6], Margaux Leemans[1,2], Susanne Terheyden[3,7], Arsen Petrovic[8], Rodrigo Gallardo [9,10], Laura M. Nederveen-Schippers[3], Panagiotis S. Athanasopoulos[3], Henderikus Pots[3], Peter J.M. Van Haastert[3], Frank Sobott[6,11,12], Christian Johannes Gloeckner [4,5], Rouslan Efremov[1,2], Arjan Kortholt[3] & Wim Versées[1,2]

Mutations in LRRK2 are a common cause of genetic Parkinson's disease (PD). LRRK2 is a multi-domain Roco protein, harbouring kinase and GTPase activity. In analogy with a bacterial homologue, LRRK2 was proposed to act as a GTPase activated by dimerization (GAD), while recent reports suggest LRRK2 to exist under a monomeric and dimeric form in vivo. It is however unknown how LRRK2 oligomerization is regulated. Here, we show that oligomerization of a homologous bacterial Roco protein depends on the nucleotide load. The protein is mainly dimeric in the nucleotide-free and GDP-bound states, while it forms monomers upon GTP binding, leading to a monomer-dimer cycle during GTP hydrolysis. An analogue of a PD-associated mutation stabilizes the dimer and decreases the GTPase activity. This work thus provides insights into the conformational cycle of Roco proteins and suggests a link between oligomerization and disease-associated mutations in LRRK2.

[1] VIB-VUB Center for Structural Biology, Pleinlaan 2, 1050 Brussels, Belgium. [2] Structural Biology Brussels, Vrije Universiteit Brussel, Pleinlaan 2, 1050 Brussels, Belgium. [3] Department of Cell Biochemistry, University of Groningen, Groningen 9747 AG, The Netherlands. [4] German Center for Neurodegenerative Diseases (DZNE), 72076 Tübingen, Germany. [5] Eberhard Karls University, Institute for Ophthalmic Research, Center for Ophthalmology, 72076 Tübingen, Germany. [6] Department of Chemistry, Biomolecular & Analytical Mass Spectrometry group, University of Antwerp, 2020 Antwerp, Belgium. [7] Structural Biology Group, Max-Planck Institute of Molecular Physiology, 44227 Dortmund, Germany. [8] Department of Mechanistic Cell Biology, Max-Planck Institute of Molecular Physiology, 44227 Dortmund, Germany. [9] VIB Center for Brain & Disease Research, 3000 Leuven, Belgium. [10] Switch Laboratory, Department of Cellular and Molecular Medicine, KU Leuven, Herestraat 49, PB 802, 3000 Leuven, Belgium. [11] Astbury Centre for Structural Molecular Biology, University of Leeds, LS2 9JT Leeds, UK. [12] School of Molecular and Cellular Biology, University of Leeds, LS2 9JT Leeds, UK. Egon Deyaert and Lina Wauters contributed equally to this work. Correspondence and requests for materials should be addressed to W.Vée. (email: wim.versees@vib-vub.be)

Mutations in the gene encoding leucine-rich-repeat kinase 2 (LRRK2) are the most common genetic cause of Parkinson's disease (PD)[1, 2]. LRRK2 mutations account for 5–6% of familial PD cases, and are identified as a risk factor for sporadic forms of the disease[3]. LRRK2 is a large (2527 amino acids) multi-domain protein, belonging to the Roco protein family. This protein family is characterized by the presence of a GTPase domain, called Roc (Ras of complex proteins), fused to a COR domain (C-terminal of Roc)[4, 5]. In many cases this Roc-COR module is preceded by a leucine-rich-repeat (LRR) domain and followed by a protein kinase domain. In LRRK2 a number of additional protein-protein interaction domains are present[6, 7]. The pathogenic LRRK2 mutations mainly cluster in the catalytic Roc-COR and kinase domains, and the most prevalent mutations result in decreased GTPase activity and/or enhanced kinase activity[8–13]. This coupled phenotype may point towards an intramolecular regulatory mechanism between the Roc and kinase domains, thus underscoring the central role of the Roc GTPase cycle in PD pathology.

Significant progress in our understanding of the structure and mechanism of the Roc-COR module of LRRK2 comes from studies with related Roco proteins from prokaryotes and lower eukaryotes[7, 14–16]. Most importantly, the model that LRRK2 functions as a GAD (G protein activated by nucleotide-dependent dimerization) is heavily based on the crystal structure of the dimeric Roc-COR module of the Roco protein from *Chlorobium tepidum*[14, 17, 18]. This model implies that the COR domain acts as a permanent dimerization device and that the stimulation of GTPase activity depends on reciprocal complementation of two Roc active sites[7, 14, 15, 17, 19]. Similar to the prokaryotic Roco proteins, various studies report that LRRK2 can also form dimers through its Roc-COR domain in vitro[15, 20–22], although some other studies suggest that the protein is mainly monomeric[23]. A number of recent results indicate that in vivo functional LRRK2 cycles between a predominantly monomeric kinase-inactive form in the cytosol and a dimeric kinase-active form at the plasma membrane[24, 25]. However, so far the mechanisms regulating these changes in LRRK2 translocation and oligomerization are poorly understood.

Here, we show that Roco proteins cycle between a monomeric and dimeric form during GTP turnover. Using small angle X-ray scattering (SAXS), multi-angle light scattering (MALS), native mass spectrometry (MS), analytical ultracentrifugation (AUC) and electron microscopy (EM), we demonstrate that the *C. tepidum* Roco protein is mainly dimeric in the nucleotide-free and GDP-bound states, while it is mainly monomeric when bound to GTP. Moreover, using time resolved Förster resonance energy transfer (FRET) and EM we show that the GTP-induced mono-merization occurs on a catalytically relevant time scale and that the monomer-dimer cycle occurs concomitant with GTP turnover. A mutation linked to PD decreases the GTPase activity by interfering with the monomer-dimer equilibrium. Together these results shed new light on a long-standing discussion regarding the oligomeric state of Roco proteins, and propose a model for their GTP hydrolysis mechanism.

## Results

**GppNHp binding induces monomerization of the Roc-COR module.** Like most prokaryotic Roco proteins, the Roco protein from *Chlorobium tepidum* (CtRoco) consists of an N-terminal LRR domain (a.a. 1–411), a central Roc-COR module (a.a. 412–946) and a C-terminal region of unknown structure and function (a.a. 947–1102) (Supplementary Fig. 1)[4, 14, 15]. A crystal structure of the Roc-COR domain construct of CtRoco (CtRoc-COR) was solved, with the Roc GTPase domain in a nucleotide-

free state[14]. This structure shows the protein as a homodimer, for which most of the contacts between the subunits are mediated via COR domain residues. To investigate the influence of nucleotide binding on the conformation of CtRoc-COR we set out to perform small angle X-ray scattering coupled to size-exclusion chromatography (SEC-SAXS) experiments with the protein in either the nucleotide-free state or saturated with GDP or the GTP mimic 5′-guanylyl imidodiphosphate (GppNHp) (Supplementary Fig. 2). The scattering profile of the nucleotide-free protein is in excellent agreement with the symmetrical CtRoc-COR dimer of the crystal structure (pdb 3DPU, Gotthardt et al. 2008), and a comparison of the experimental and theoretical scattering profiles yields a $\chi^2$-value of 0.9 (Fig. 1a). Moreover, a good agreement is found between the *ab initio* molecular envelope based on the SAXS profile and the CtRoc-COR crystal structure (Fig. 1b). However, when CtRoc-COR is saturated with either GDP or GppNHp conformational changes are taking place, as the experimental scattering curves yield a worse fit with the theoretical scattering curve based on the nucleotide-free crystal structure (translated in a $\chi^2 = 1.6$ and $\chi^2 = 6.0$ for GDP- and GppNHp-bound protein respectively). To obtain further insights in the nature of these conformational changes we calculated the estimated molecular masses based on the Porod volumes. For both the nucleotide-free and the GDP-bound form, a molecular mass around 120 kDa is found, which is in good agreement with the expected molecular mass of the dimer (theoretical $MM_{dimer} = 130$ kDa) (Fig. 1c). In the GppNHp-bound state an average molecular mass of 90 kDa is found, in between the values expected for a monomer and a dimer. This indicates a monomer/dimer equilibrium with overlapping peaks in the chromatogram at the concentration used in SEC-SAXS (50 µl of an 8 mg ml$^{-1}$ protein solution injected on the column). Such a shift in oligomerization is also translated in the pair-distance distribution functions, where an overlay shows that the curves overlap for the nucleotide-free and GDP loaded protein, while a shift toward on average smaller distances is found for the GppNHp-bound protein (Supplementary Fig. 2d).

Next, we performed multi-angle light scattering coupled to size-exclusion chromatography (SEC-MALS) and sedimentation velocity analytical ultracentrifugation (SV-AUC) experiments to determine the molecular mass of CtRoc-COR in different nucleotide states (Figs. 1c, d). For the SEC-MALS experiments, 10 µl of 8 mg ml$^{-1}$ protein solutions were injected on the SEC column. For the nucleotide-free and GDP-bound forms a molecular mass of 120 and 125 kDa is obtained respectively, again corresponding to a dimer. For the GppNHp-loaded protein SEC-MALS yielded a molecular mass of 78 kDa, just slightly above the expected value of a monomer. The SV-AUC experiments were performed at an even lower protein concentration (0.3 mg ml$^{-1}$, 4.6 µM) (Fig. 1c and Supplementary Fig. 3). Again, for nucleotide-free and GDP-loaded CtRoc-COR molecular masses of 121 and 115 kDa are obtained (sedimentation coefficients of 4.7 S and 4.8 S, respectively). In the presence of GppNHp the monomer-dimer equilibrium is almost completely shifted to the monomer, clearly seen as a shift in sedimentation coefficient from around 4.8 S (dimer) to 3.6 S (monomer), corresponding to a molecular mass of 69 kDa.

Finally, to resolve monomeric and dimeric species we performed native MS experiments (Fig. 1e). The mass spectrum of nucleotide-free CtRoc-COR shows exclusive dimeric species ($MM_{exp} = 130$ kDa). Also with GDP dimeric species dominate, while a very small amount of monomer appears ($MM_{exp} = 65$ kDa). However, in agreement with our previous data, we observe a clear shift toward the monomeric species in the presence of GppNHp, resulting in an approximately 50:50 monomer/dimer ratio under the conditions used in this experiment.

In conclusion, all these experiments show that while the CtRoc-COR is mainly a dimer in the nucleotide-free and GDP-bound form, the GTP mimic GppNHp induces monomerization in a concentration dependent manner.

**GppNHp shifts the dimer-monomer equilibrium in CtRoco.** Considering the GppNHp-induced monomerization observed in the CtRoc-COR construct, we next collected SAXS data for the full length CtRoco protein, either in the absence of nucleotides or bound to GDP or GppNHp (Supplementary Fig. 4). As there is, thus far, no crystal structure available of the full length CtRoco

protein (nor of any other Roco protein), no fitting of the SAXS data on a theoretical curve could be performed. However, visual inspection of the scattering curves clearly shows that conformational changes occur in both the CtRoco GppNHp- and GDP-bound states compared to the nucleotide-free state (Fig. 2a and Supplementary Fig. 4e). Docking of the CtRoc-COR crystal structure into the *ab initio* envelope generated from the SAXS data of the nucleotide-free CtRoco protein show additional features at the N- and C-terminus of CtRoc-COR presumably corresponding to the LRR and C-terminal domains (Fig. 2b).

Molecular mass calculation from the Porod analysis of the SAXS curves, from SEC-MALS and from SV-AUC (Figs. 2c, d

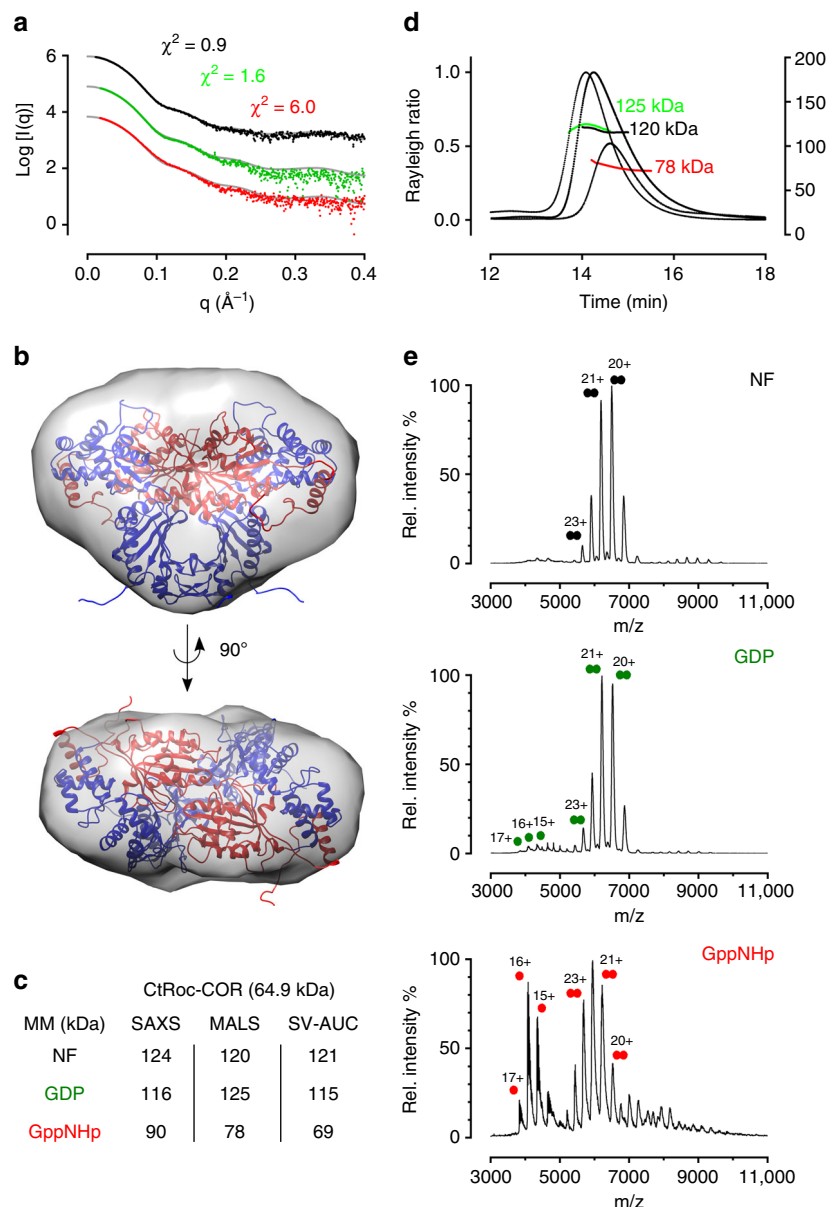

**c**

CtRoc-COR (64.9 kDa)

| MM (kDa) | SAXS | MALS | SV-AUC |
|---|---|---|---|
| NF | 124 | 120 | 121 |
| GDP | 116 | 125 | 115 |
| GppNHp | 90 | 78 | 69 |

**Fig. 1** The *Chlorobium tepidum* Roc-COR domain monomerizes upon binding of a non-hydrolysable GTP analogue (GppNHp). **a** Pairwise-comparison using CRYSOL of the theoretical scattering curve derived from the crystallographic dimer model of CtRoc-COR (PDB 3DPU, *grey line*) with the experimental scattering curves of CtRoc-COR in the absence of nucleotides (*black dots*) or in the presence of GDP (*green dots*) or GppNHp (*red dots*). **b** Superposition of the crystallographic CtRoc-COR dimer model (Roc in *red* and COR in *blue*) on the *ab initio* SAXS envelope, constructed starting from the scattering curve of CtRoc-COR in the absence of nucleotides. **c** Overview of the molecular masses obtained via SAXS (based on Porod volume/1.7), SEC-MALS and SV-AUC for CtRoc-COR in different nucleotide-bound states. The theoretical molecular mass of the monomer is given in between brackets. (NF = nucleotide-free) **d** SEC-MALS data for CtRoc-COR in the absence (*black*) or presence of nucleotides GDP (*green*) or GppNHp (*red*). **e** Native mass spectra of CtRoc-COR for the three different nucleotide states: nucleotide-free (NF), GDP-bound and GppNHp-bound. Peaks corresponding to dimeric and monomeric species are labeled with two *circles* and one *circle*, respectively

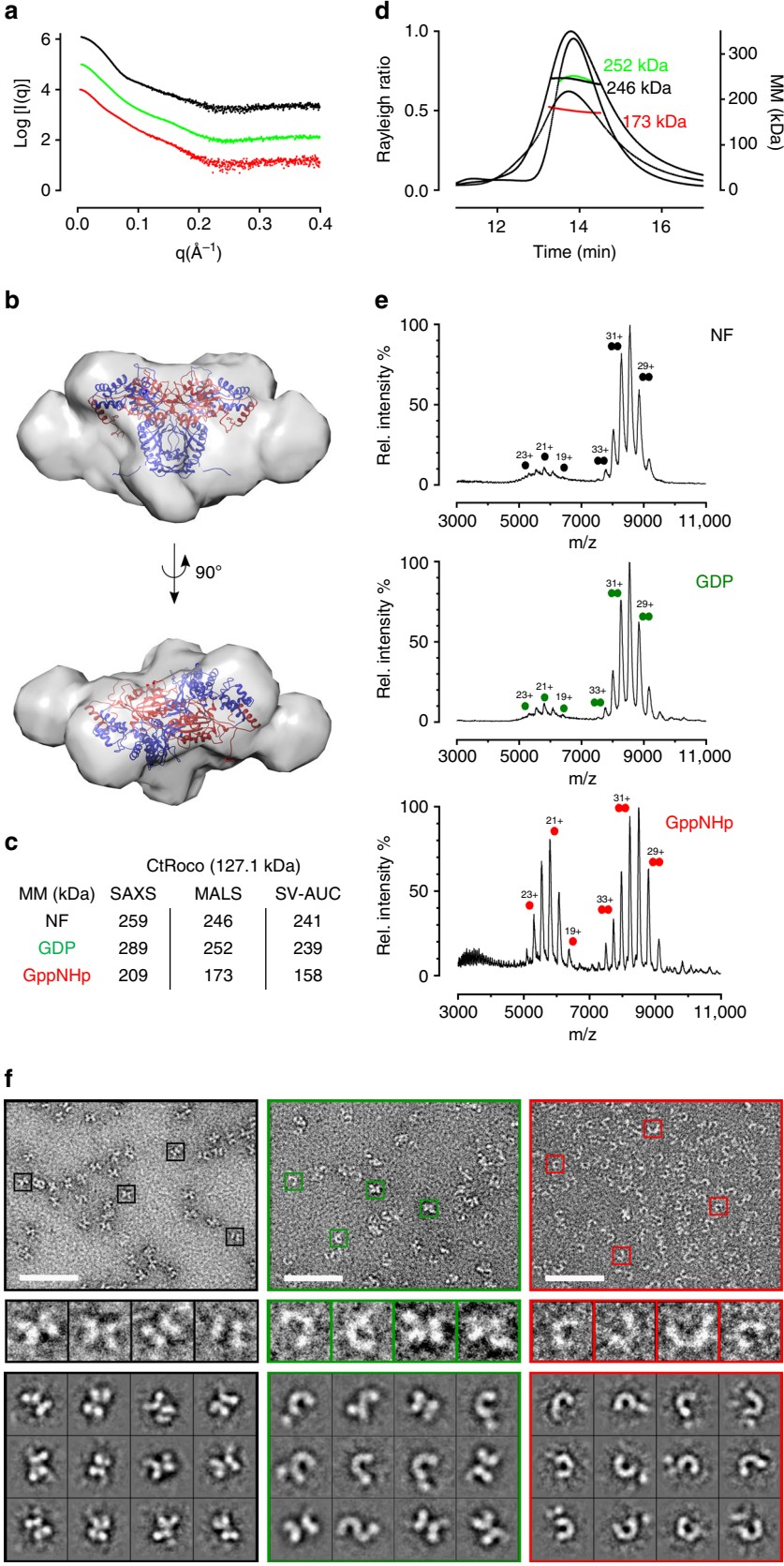

and Supplementary Fig. 5), yields values between 240 and 290 kDa for the nucleotide-free and GDP-bound forms, close to the theoretical value expected for a dimer (theoretical $MM_{dimer}$ = 254.2 kDa). In the presence of GppNHp, SEC-SAXS (50 µl of an 8 mg ml$^{-1}$ protein solution injected on a column) gives a molecular mass of 209 kDa and SEC-MALS (10 µl of an 8 mg ml$^{-1}$ protein solution injected on a column) a molecular mass of 173 kDa, while SV-AUC (protein at 0.6 mg ml$^{-1}$, 4.5 µM) gives a broader distribution with a molecular mass at the peak of 158 kDa (sedimentation coefficient of 4.9 S) (Supplementary Fig. 5). As for CtRoc-COR this indicates that GppNHp induces monomerization, with the protein existing in a concentration dependent monomer/dimer equilibrium. Finally, native-MS spectra of CtRoco show mainly dimeric species ($MM_{exp}$ = 256 kDa) with a very small amount of monomers ($MM_{exp}$ = 128 kDa) for nucleotide-free and GDP-loaded protein and a 50:50 monomer-dimer ratio for the GppNHp loaded protein (Fig. 2e).

It might seem remarkable at first sight that despite the difference in molecular mass the nucleotide-free, GppNHp- and GDP-bound CtRoco proteins elute at approximately the same volume in SEC (Fig. 2d). Correspondingly, Guinier analysis of the SAXS data shows that the radius of gyration ($R_g$) even increases upon going from the nucleotide-free state, over the GDP-bound state to the GppNHp-bound state (Supplementary Fig. 4). This could be explained by a conformational change occurring in the CtRoco subunits upon monomerization or upon nucleotide binding. Indeed, as is the case for the CtRoc-COR construct, the SAXS pair-distance distribution function of CtRoco shows a shift toward shorter average inter-atom distances in the presence of GppNHp compared to GDP or nucleotide-free states, indicative of monomerization (Supplementary Fig. 4d). However, the curve of the GppNHp-bound CtRoco protein also shows a tailing toward longer maximal distances. This strongly suggests the occurrence of a GppNHp-bound monomer that is more elongated than the corresponding subunits in the nucleotide-free dimer. Also in the GDP-bound state tailing of the pair-distance distribution function is observed indicating that already some conformational changes occur in the dimeric nucleotide-bound protein. These conformational changes also explain why the observed monomer/dimer equilibrium would probably be missed when assessed only with size-exclusion chromatography.

Since we observe a monomer/dimer equilibrium in nucleotide-bound CtRoco, we next determined the dissociation constant of this equilibrium under the different nucleotide conditions, using sedimentation equilibrium AUC (SE-AUC) experiments (Supplementary Fig. 6). Global analysis of the data using a single species model indicated predominantly dimeric species for the nucleotide-free (239 kDa) and the GDP-bound CtRoco (225 kDa), whereas a clear shift to the monomeric state for GppNHp-bound CtRoco (177 kDa) is observed. Fitting the data for GDP- and GppNHp-bound CtRoco using a monomer/dimer model

results in an approximate $K_D$ of 2 µM for GDP-bound CtRoco and 30 µM for GppNHp-bound CtRoco (exclusively dimers are found for nucleotide-free CtRoco) (Supplementary Fig. 6). Since protein concentrations used for the SEC-SAXS, SEC-MALS, SV-AUC and native MS experiments range between approximately 4.5 and 16 µM, these dissociation constants generally agree with the predominantly dimeric form observed for CtRoco bound to GDP and the shift to the monomeric form upon GppNHp binding.

Finally, to further characterize the observed nucleotide-induced changes in oligomerization we turned to negative stain electron microscopy (EM). Thereto, CtRoco was applied on grids at a concentration of 0.01 mg ml$^{-1}$ (0.08 µM) either in a nucleotide-free state or pre-incubated with 1 mM GDP or GppNHp (Fig. 2f and Supplementary Fig. 7). In the nucleotide-free state a rather uniform population of "X-shaped" particles is observed, displaying seemingly a two-fold symmetry. We interpret these particles as being the dimeric form of CtRoco. In contrast, in the presence of GppNHp a uniform population of more elongated "worm-like" particles is observed lacking the two-fold symmetry. We interpret these particles to correspond to the monomeric form of CtRoco, in agreement with the SAXS, SEC-MALS, native MS and AUC experiments. For the GDP-bound state a mixture of particles corresponding to dimers and monomers is observed.

In conclusion, we show that nucleotide binding also induces monomerization of CtRoco in a concentration dependent manner, with the GTP mimic GppNHp having a much stronger effect than GDP.

**Monomerization occurs on a relevant time scale.** In order to be relevant for the CtRoco-catalysed GTP hydrolysis reaction, the observed monomerization should occur within a time frame that is consistent with the time of a full GTP turnover cycle. The $k_{cat}$ value for GTP hydrolysis of CtRoco and CtRoc-COR is 0.1 min$^{-1}$ (Supplementary Fig. 8), meaning that it takes a protein molecule, on average, approximately 10 min to travel through an entire GTPase cycle under conditions of substrate saturation. For the CtRoc-COR construct the rate of monomerization could be measured using time-resolved FRET experiments making use of a site-specific single cysteine mutant of this protein construct[19]. A CtRoc-COR variant containing a single cysteine residue at position 928 in the COR domain (S928C) was randomly labelled with a Cy3/Cy5 FRET pair using maleimide-chemistry (Fig. 3a). Statistical incorporation should result in 50% incorporation with the donor/acceptor pair. Subsequently, this nucleotide-free protein was rapidly mixed in a stopped flow apparatus with either buffer or 50 µM GDP, GppNHp or GTP, and the decrease in FRET signal linked to monomerization was followed (Fig. 3b). Compared to the buffer control, a very small and slow decrease in FRET signal is observed with GDP. In contrast, a fast decrease in

**Fig. 2** Binding of a non-hydrolysable GTP analogue (GppNHp) to the *Chlorobium tepidum* Roco protein shifts the equilibrium toward the monomeric form. **a** Scattering *curves* of CtRoco in the absence (*black dots*) or presence of nucleotides GDP (*green dots*) or GppNHp (*red dots*). **b** Superposition of the crystallographic CtRoc-COR dimer model (Roc in *red* and COR in *blue*) onto the *ab initio* SAXS envelope, constructed starting from the scattering curve of CtRoco in the absence of nucleotides. The envelope shows clear additional features corresponding to the N-terminal LRR domain and the C-terminal domain, which are not present in the CtRoc-COR crystal structure. **c** Overview of the molecular masses obtained via SAXS (based on Porod volume/1.7), SEC-MALS and SV-AUC for CtRoco in different nucleotide-bound states. The theoretical molecular mass of the monomer is given in between brackets. (NF = nucleotide-free) **d** SEC-MALS data for CtRoco in the absence (*black*) or presence of nucleotides GDP (*green*) or GppNHp (*red*). **e** Native mass spectra of CtRoco for the three different nucleotide states: nucleotide-free (NF), GDP-bound and GppNHp-bound. Peaks corresponding to dimeric and monomeric species are labeled with two *circles* and one *circle*, respectively. **f** *Top panels*: negative-stain EM images (*scale bar*: 50 nm) of CtRoco in the nucleotide-free state (*black border*), bound to GDP (*green border*) or bound to GppNHp (*red border*). *Middle panels*: 4x enlargements of boxed particles in top panels (from *left* to *right*). *Bottom panels*: representative class averages (box size: 18.2 × 18.2 nm) for CtRoco in nucleotide-free state (*black border*), or bound to GDP (*green border*) or GppNHp (*red border*). See also Supplementary Fig. 7 for an overview of all class averages

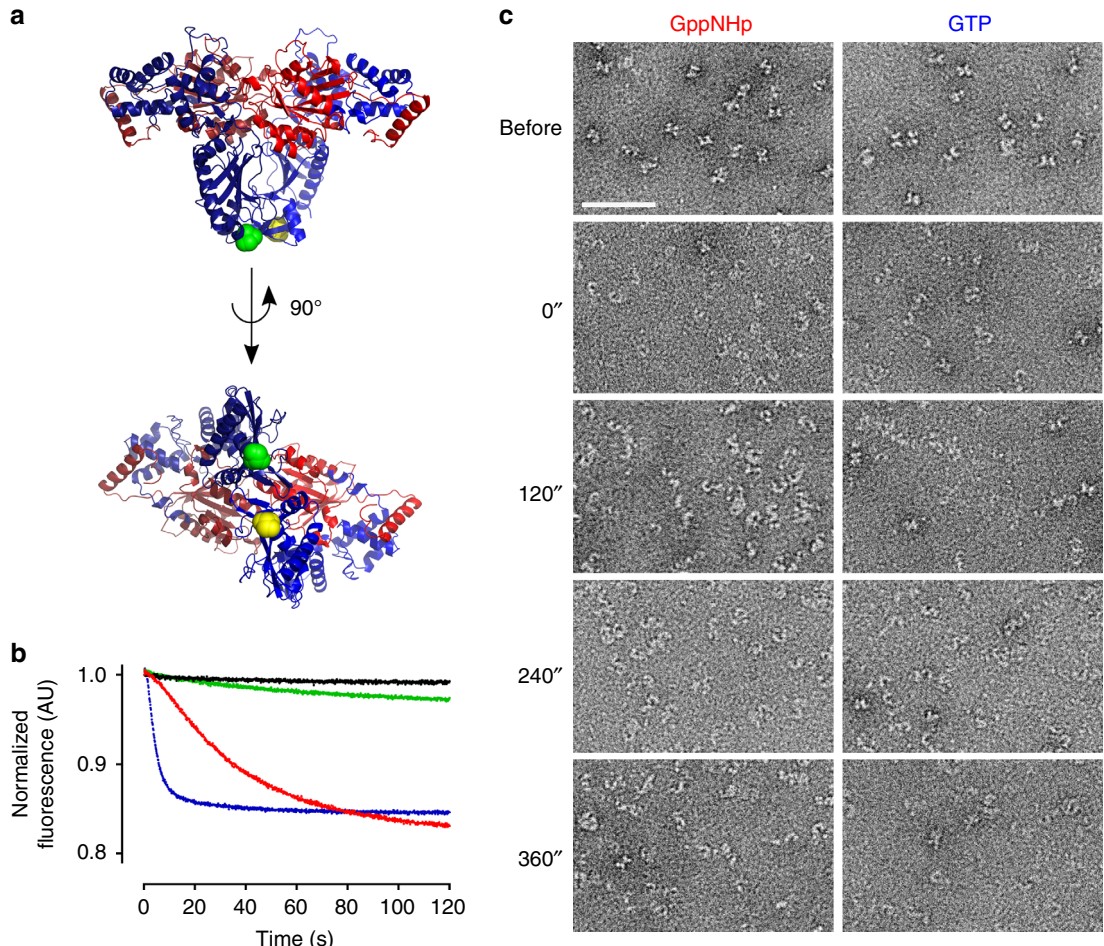

**Fig. 3** GppNHp- and GTP-induced monomerization of the *Chlorobium tepidum* Roco protein occurs in a time scale relevant for GTP turnover. **a** Two different views of the dimeric CtRoc-COR crystal structure with amino acid 928 colored *green* and *yellow* for protomer A and B, respectively. Within the dimeric protein these residues are separated by a distance of 20 Å allowing them to be used for Cy3/Cy5 FRET experiments. The Roc and COR domains are colored *red* and *blue* (dark for protomer A and light for protomer B), respectively. **b** Stopped-flow FRET traces of labelled CtRoc-COR upon mixing with different nucleotides to follow monomerization in time. The FRET signal of Cy3/Cy5-labeled CtRoc-COR (S928C) over time is shown after rapid mixing with buffer (*black*), or 50 μM GDP (*green*), GppNHp (*red*) or GTP (*blue*). **c** Monomerization of CtRoco followed via time-resolved negative stain EM. CtRoco was mixed with 1 mM of GppNHp or GTP and samples were taken every 2 min. Representative EM images of the samples just before adding nucleotide and for each time point are shown. (*scale bar*: 50 nm)

FRET signal is observed upon mixing the protein with an excess of either GppNHp or GTP. The decrease in signal and thus the monomerization is significantly faster upon mixing with GTP compared to the GTP mimic GppNHp. While monomerization reaches a steady state within less than 25 s with GTP, equilibrium is only reached after more than 100 s with GppNHp. Such differences between physiologically relevant nucleotides and nucleotide analogues were described before[26]. In any case, the observed time for monomerization is significantly faster than the GTP turnover time of about 10 min meaning that monomerization happens on a catalytically relevant time scale.

Since cysteine-free full length CtRoco could not be obtained, we turned to time-resolved EM measurements to estimate the time frame of monomerization. Hereto, CtRoco was mixed with an excess of GppNHp or GTP and samples were taken every two minutes and immediately spotted on grids. In agreement with the FRET experiments nearly full conversion of CtRoco from the "X-shaped" dimers to the "worm-shaped" monomers is observed at the 2 min time point (Fig. 3c), while a significant amount of monomerization has even taken place within the dead time (2 min) of the experiment. These experiments thus confirm that

GTP-induced monomerization of CtRoco is fast and occurs within the time frame of a catalytic cycle of GTP turnover.

**CtRoco completes a monomer-dimer cycle during GTP turnover.** Considering that GTP-induced formation of CtRoco/CtRoc-COR monomers is fast compared to a complete GTP turnover, we subsequently assessed whether the protein undergoes a monomer-dimer cycle coupled to GTP turnover. We therefore resorted to single turnover kinetic measurements (1 μM protein + 1 μM GTP), such that a complete single GTPase cycle can be monitored.

Reversed-phase HPLC measurements show that, under the experimental (non-saturating) conditions, it takes about 6000 s for 1 μM CtRoc-COR to completely convert 1 μM GTP to GDP (Fig. 4a). In turn, stopped-flow measurements where 1 μM of fluorescent 2′-(or-3′)-O-(N-methylanthraniloyl)-GTP (mant-GTP) is mixed with 1 μM of CtRoc-COR show a fast increase in fluorescence, coupled to mant-GTP binding and the associated conformational changes, that occurs in a time frame of about 250 s (Fig. 4a). This fast binding phase is followed by a slow decrease

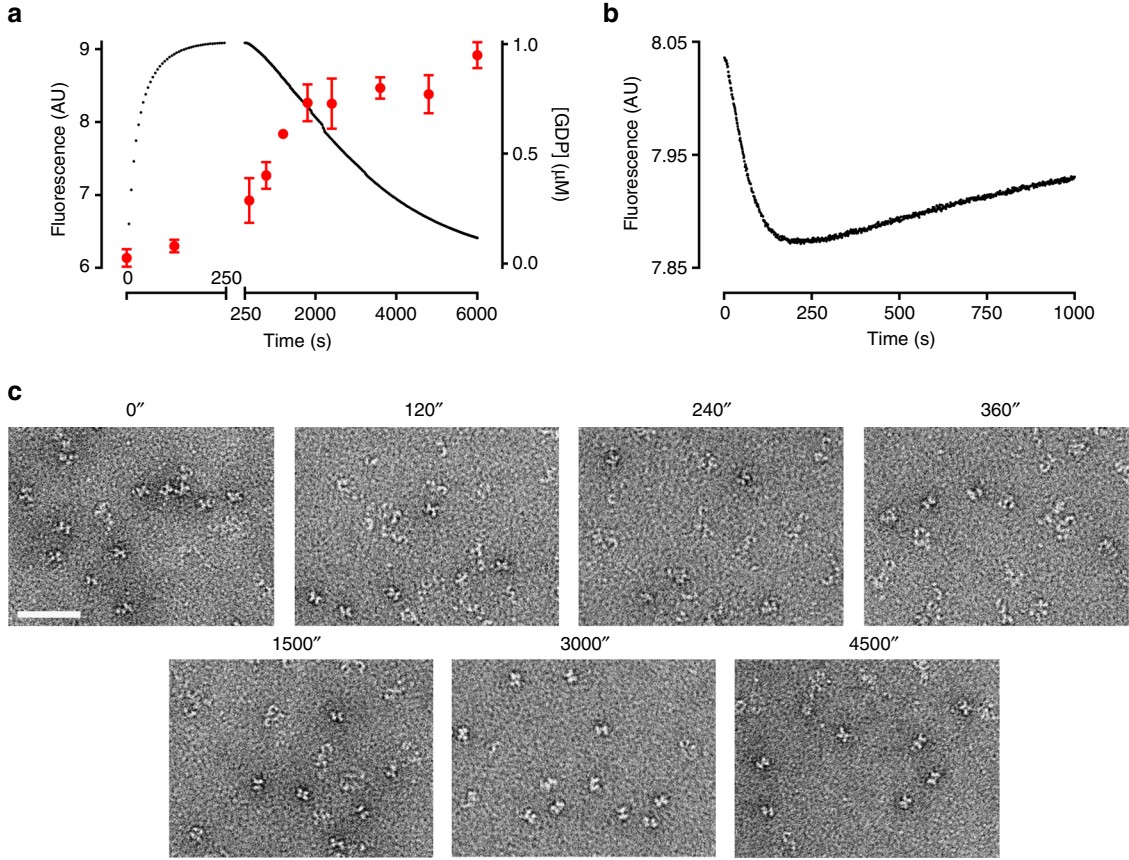

**Fig. 4** The *Chlorobium tepidum* Roco protein undergoes a monomer-dimer cycle during GTP turnover. **a** Single turnover (mant-)GTP hydrolysis by CtRoc-COR (S928C) followed by stopped flow fluorescence (*black curve*) and reversed-phase HPLC (*red* data points; each data point is the average (±s.d.) of 3 independent measurements). Rapid mixing of 1 μM mant-GTP and 1 μM unlabeled CtRoc-COR (S928C) in a stopped-flow apparatus yields a rapid increase in fluorescence (time frame 1–250 s), followed by a slow decrease in a time frame of 6000 s. Following production of GDP from GTP in time via reversed-phase HPLC shows that the increase in fluorescence occurs prior to GDP production while all GTP is converted in the time scale of 6000 s. A split time axis is used to highlight the two phases: fast fluorescence increase associated with GTP binding and slow fluorescence decrease concomitant with GTP hydrolysis. **b** Stopped-flow FRET signal obtained by mixing 1 μM of Cy3/Cy5- labeled CtRoc-COR (S928C) with 1 μM GTP. The traces show relatively fast monomerization (first phase from 1–250 s) followed by a slow return to the dimeric state after GTP hydrolysis. **c** Single GTP turnover of CtRoco followed by time resolved EM. 1 μM CtRoco was mixed with 1 μM GTP and samples were taken at the indicated time points. Representative images for each time point are shown (see also Supplementary Fig. 9) (*scale bar*: 50 nm)

in fluorescence coupled to mant-GTP hydrolysis and return to the initial conformation. In agreement with the data obtained from the reversed-phase HPLC measurements, the total GTPase cycle is finished within about 6000 s. Subsequently we followed the monomerization of CtRoc-COR during single GTP-turnover using our time-resolved FRET approach. 1 μM Cy3/Cy5-labelled CtRoc-COR (S928C) was rapidly mixed with 1 μM GTP and the FRET signal was followed over time (Fig. 4b). These traces show a fast decrease in the FRET signal associated with monomerization, followed by a slow increase in signal associated with dimerization. This shows that the GTP-induced monomerization is reversible and re-dimerization occurs upon GTP hydrolysis. The FRET signal reaches a minimum at about 250 s. This time point of maximal monomerization corresponds to the time point of maximal GTP binding as determined from the fluorescence stopped-flow experiments with mant-GTP. After this point the FRET signal increases in a time frame that corresponds to GTP hydrolysis and probably also GDP release (note that for CtRocCOR $K_D(GDP) = 30.2$ μM and $K_D(GppNHp) = 0.33$ μM, while a concentration of 1 μM is used here[14]).

Finally, we also assessed the oligomeric state of full length CtRoco during single turnover GTP hydrolysis using time-resolved negative stain EM measurements. Hereto, 1 μM of

CtRoco was mixed with 1 μM GTP and samples were taken at different time points (0, 120, 240, 360, 1500, 3000 and 4500 s) (Fig. 4c and Supplementary Fig. 9). Under these single turnover conditions a large fraction of CtRoco molecules are converted from a dimeric into a monomeric form after 240 s. This is expected for a single turnover experiment where a mixture of unbound, GTP-bound and post-hydrolysis GDP-bound CtRoco molecules co-exist. After that time point the fraction of dimeric protein is again increasing due to GTP hydrolysis, and after 4500 s, when all GTP is hydrolysed, a dimeric population reforms.

Thus, together these experiments clearly show that during the GTPase cycle, CtRoco undergoes monomerization concomitant with GTP binding and subsequent dimerization coupled to GTP hydrolysis and GDP release.

**An analogue of a PD mutation stabilizes the CtRoco dimer.** PD-associated mutations in LRRK2 are mainly located in the kinase and Roc-COR domains. Mapping of the Roc-COR mutations (LRRK2 I1371V/R1441C/Y1699C) onto the *C. tepidum* Roc-COR crystal structure showed that the bacterial analogues of these mutations (L487A/Y558A/Y804C) are all located in the conserved interface between the Roc and COR sub-domains[14].

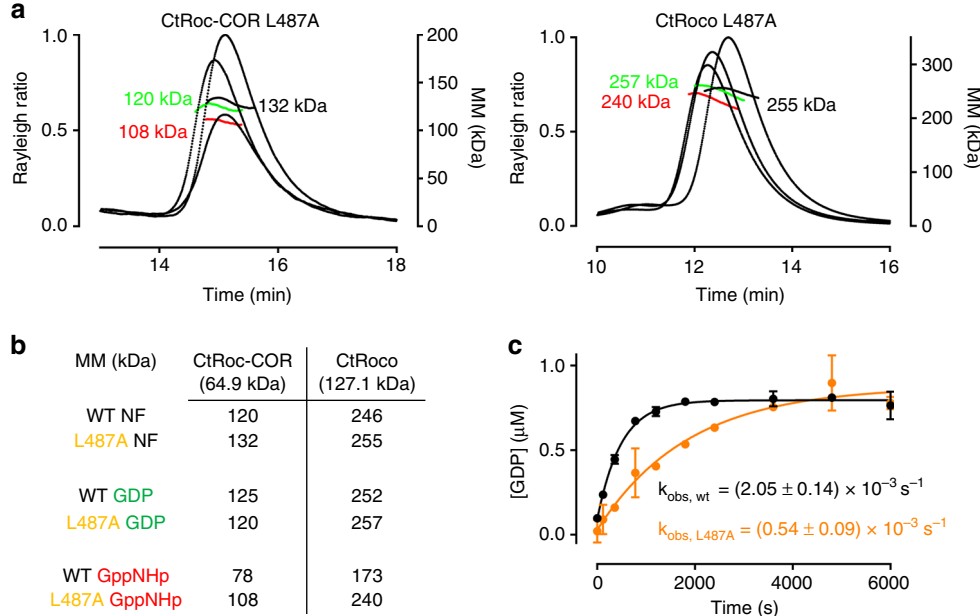

**Fig. 5** Effect of a PD-analogous mutation on the CtRoco and CtRoc-COR monomer-dimer equilibrium. **a** SEC-MALS data for the CtRoc-COR L487A mutant (*left panel*) and the CtRoco L487A mutant (*right panel*) in the absence (*black*) or presence of nucleotides GDP (*green*) or GppNHp (*red*). **b** Overview of molecular masses determined by SEC-MALS for CtRoc-COR wt, CtRoc-COR L487A, CtRoco wt and CtRoco L487A in absence and presence of GDP or GppNHp. The theoretical molecular masses of the monomers are given in between brackets. (NF = nucleotide-free). **c** Single turnover GTP hydrolysis of CtRoc-COR wt (*black*) and CtRoc-COR L487A (*orange*) using 1 μM protein and 1 μM GTP. GDP formation is followed using reversed-phase HPLC. The observed rate constants ($k_{obs}$) ± s.e. for wild-type and mutant CtRoc-COR are indicated. Each data point is the average (±s.d.) of 3 independent measurements

Since, in our hands, the Y558A and Y804C mutants show significant aggregation when analysed on size exclusion chromatography, we focused on the L487A mutant and analysed its effect on the oligomeric state of the CtRoco and CtRoc-COR proteins using SEC-MALS. Residue L487 corresponds to I1371 in LRRK2 and is located in the Roc domain, in the C-terminus of Switch 1, at the interface of the Roc and COR domains. In their nucleotide-free and GDP-bound states, both the CtRoco and CtRoc-COR proteins that contain the L487A mutation are entirely dimeric, as was also observed in the wild-type counterparts (Figs. 5a, b). However, while in the GppNHp-bound form the monomer-dimer equilibrium of the wild-type CtRoc-COR and CtRoco is clearly shifted toward the monomeric form, the corresponding proteins harboring the L487A mutation remain nearly completely dimeric (Figs. 5a, b). Thus we find that the PD-associated L487A mutation stabilizes the dimeric form of the CtRoco protein. The same behavior is observed when we compare the SAXS curves of wild-type and L487A CtRoco in different nucleotide states (Supplementary Fig. 10, 11). While the scattering curves, the normalized Kratky plots[27] and the pair distance distribution function of the wild-type and L487A proteins nearly overlap in the nucleotide-free and GDP states, the curves of both protein variants clearly differ in the GppNHp-bound forms. This again indicates that, while the L487A mutation does not cause any large-scale conformational changes in the nucleotide-free and GDP-bound dimers, it does stabilize the dimeric form of the protein in the GppNHp-bound state.

Subsequently, we measured the GTPase activity of the wild-type and L487A CtRoc-COR proteins under single-turnover conditions and find that the L487A mutation causes a 4-fold decrease in the single-turnover rate constant (Fig. 5c). This is in agreement with previous reports that show that the PD mutations in the Roc-COR domain of LRRK2, as well as the counterparts in CtRoco, decrease the GTPase activity[10, 12, 28–30]. We thus

speculate that monomerization of the Roco protein, which we showed to be an integral part of the GTPase cycle, is hindered by this PD-analogous mutation, thereby causing the pathological decrease in GTPase activity. Hence, our data present a link between the Roco monomer-dimer transition, the rate of GTP turnover and disease.

## Discussion

In the current study we show that the oligomerization of the *Chlorobium tepidum* Roco protein depends on its nucleotide state. In contrast to previous suggestions, we find that the protein is mainly monomeric when bound to GTP and dimeric in the nucleotide-free state, while an intermediate situation is observed in the GDP state. Moreover, we show that the CtRoco protein cycles through the dimeric and monomeric conformations during a round of GTP hydrolysis, meaning that these changes in oligomerization are an integral feature of the GTPase catalytic cycle. Consequently, although many details still need to be resolved, a mechanism as outlined in Fig. 6 can be proposed based on our results. The reaction cycle starts with the rapid binding of a GTP molecule to the Roc domains, followed by monomerization of the protein as observed in SEC-MALS, SEC-SAXS, native MS, AUC and EM (Figs. 1–3). Inter-domain and inter-subunit conformational changes leading to monomerization are likely triggered via changes in conformation of the switch regions. The crystal structure of CtRoc-COR shows that especially switch II is ideally positioned to transfer GTP-induced conformational changes from the Roc domain to the COR domain within the same protomer as well as to the adjacent COR domain[14]. The monomerization step is followed by slow GTP hydrolysis concomitant with protein dimerization (Fig. 4). Considering that single turnover GDP production, as monitored via reversed-phase HPLC, occurs on the same time scale as dimerization (Fig. 4), we assume GTP

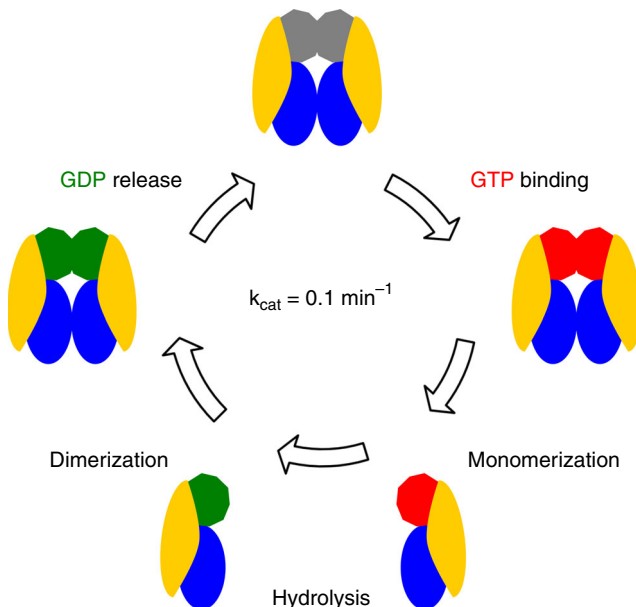

**Fig. 6** Proposed monomer-dimer transitions during the GTP hydrolysis cycle of CtRoco. In its nucleotide-free state the protein is dimeric. GTP binding induces conformational changes that lead to fast monomerization. In the monomeric state the protein hydrolyses GTP to GDP. Depending on the protein concentration, protein dimerization could take place either after or before release of GDP. After GDP is released, the cycle can restart. In the cartoon of CtRoco, the LRR domain is coloured *yellow*, the COR domain *blue*, the Roc domain dependent on the nucleotide state grey (nucleotide-free), *green* (GDP-bound) or *red* (GTP-bound)

hydrolysis to be the rate-limiting step in this process. Finally, GDP is released, which may occur either after or concomitant with protein dimerization since we find CtRoco to be in equilibrium between a monomeric and dimeric state in the presence of GDP.

Our observation that GTP hydrolysis occurs after monomerization contrasts with the current model that assumes that Roco proteins, including LRRK2, function as GADs[17, 18, 31]. This latter model is based largely on the crystal structure of the nucleotide-free Roc-COR module of the *C. tepidum* Roco protein, being the only available structure of a full Roc-COR tandem domain[14]. This structure shows a Roc-COR dimer where the dimer contacts are formed by the COR domain and with the GTP binding sites of the Roc domains juxtaposed. This observation has led to the hypothesis that GTP binding induces dimerization of the Roc domains, with the reciprocal complementation of the active sites finally leading to catalysis of GTP hydrolysis. We show here, using the exact same protein, that GTP binding induces monomerization and that GTP hydrolysis occurs after the monomerization step. So far, it remains unclear whether Roco monomerization is a strict requirement for GTP hydrolysis to occur or whether GTP-induced monomerization and GTP hydrolysis are two uncoupled phenomena. However, both EM and SAXS indicate the occurrence of secondary inter-domain conformational changes in the CtRoco monomers compared to the dimers, with the protomers in the monomeric arrangement being more elongated compared to the dimers (Fig. 2 and Supplementary Fig. 4d). High resolution structural information of the monomeric states could shed further light on the mechanism of GTP hydrolysis. A second question that is triggered by our findings concerns the role of dimerization within the reaction cycle. A potential scenario would be that dimerization of the Roco subunits leads to a decrease in the affinity for GDP, thus enhancing the rate of GDP release.

Since the presence of the Roc-COR module is highly conserved throughout all Roco proteins, our current findings also throw new light on LRRK2 functioning. However, apart from the Roc-COR and LRR domain, LRRK2 also contains a kinase domain and a number of domains typically involved in protein-protein interactions, which potentially can have an additional influence on oligomerization[21, 32]. Detailed and quantitative biophysical studies with LRRK2 to determine its oligomeric state remain highly challenging since purification yields are low, and purified LRRK2 is very heterogeneous and prone to degradation. However, it has been shown that LRRK2 can form dimers via its Roc-COR domain[15, 20–22], while other studies indicate that LRRK2 can also exist as a monomer[18, 20, 23]. Both observations can be reconciled with our model. Interestingly, recent data suggest that LRRK2 can exist as either a monomer or a dimer in the cell, where the monomeric form prevails in the cytosol and the dimeric form is more prominently present at the membrane[21, 24, 25, 33]. A higher prevalence of a monomeric GTP-bound form under multiple GTP turnover conditions such as present in the cytosol (GTP concentrations in the cell are in a range of 1–1.6 mM[34, 35]) corresponds to our in vitro observation in EM (Fig. 3). Following the same argument, LRRK2 would exist in a more GDP-bound form at the membrane. Apart from these considerations it is likely that additional regulatory factors, such as phosphorylation or interaction with other proteins, influence the monomer-dimer equilibrium of LRRK2 in vivo[36].

A number of point mutations in the Roc (R1441C/G/H) and COR (Y1699C) domains of LRRK2 are directly linked to PD[1, 37, 38]. These mutations have been related to a reduced GTPase activity[10, 12, 28–30], but so far no widely-accepted consensus on the effect of PD mutations on homodimerization exists[22, 29, 33, 39]. According to our model, whereby the CtRoco protein needs to cycle through a monomeric and dimeric state, stabilization of either could lead to a reduced GTPase activity. We show that the L487A mutation in the Roc-COR interface (analogous to I1371V in LRRK2) decreases the GTPase activity by stabilizing the dimer interface (Fig. 5). Our results thus provide a direct link between oligomerization, GTPase activity and disease phenotype.

Because the kinase activity of LRRK2 is directly responsible for neuronal toxicity[9, 40], most efforts for therapeutic intervention are focussed on the development of kinase-inhibitors[41, 42]. However, long-term inhibition of LRRK2 by many of these inhibitors increases the risk for morphological changes in lungs, similar to what has been observed in LRRK2 knock-out models[43–45]. Our findings now show that either increased dimerization or increased monomerization could lead to changes in GTPase activity and might form an appealing target for future drug development.

## Methods

**Protein expression and purification**. The DNA fragments coding for CtRoco (a.a. 1-1102) and CtRoc-COR (a.a. 412-946) were cloned in the pProEX plasmid with N-terminal His-tags[14]. The plasmids were transformed in *E. coli* BL21 (DE3) cells. Cells were grown at 37 °C in Terrific Broth medium with 100 μg ml$^{-1}$ ampicillin. Once an optical density at 600 nm (OD$_{600}$) of 0.7 was reached, 0.1 mM β-D-1-thiogalactopyranoside (IPTG) was added and the temperature was dropped to 28 °C. After overnight incubation, the cells were harvested by centrifugation and resuspended in buffer A (20 mM Hepes pH7.5, 150 mM NaCl, 5 mM MgCl$_2$, 5% glycerol, 1 mM DTT) with 50 μg ml$^{-1}$ DNAse, 0.1 mg ml$^{-1}$ AEBSF, 1 μg ml$^{-1}$ leupeptin, and 200 μM GDP. Afterwards the cells were disrupted using a cell disruptor (Constant systems) and the soluble fraction was separated from the cell debris via centrifugation. As a first purification step, an immobilized metal affinity chromatography (IMAC) step was performed. The supernatant containing the soluble protein was loaded on Ni$^{+2}$-NTA sepharose resin. The resin was extensively washed with buffer A and subsequently the protein was eluted using buffer A including 300 mM imidazole. The obtained protein was subsequently dialysed against 20 mM Hepes pH 7.5, 150 mM NaCl, 5% glycerol, 1 mM DTT (buffer B). Before the final size exclusion chromatography purification step, 1 mM EDTA was added to the sample and the mixture was incubated for 1 h at 4 °C. Finally the

sample was loaded on a Superdex S200 26/60 column (GE Healthcare) equilibrated with buffer B. Fractions containing the protein were supplemented with 5 mM MgCl₂ and the nucleotide-load was determined using reversed-phase chromatography coupled to HPLC (see further).

**SAXS and MALS experiments.** SAXS experiments were performed on the BM29 beamline at the ESRF (Grenoble, France) and the SWING beamline at SOLEIL (Paris, France), always with an inline HPLC set-up. 50 μl of an 8 mg ml⁻¹ protein sample was injected on a Bio SEC-3 HPLC column (Agilent, 3 μm 300 Å) equilibrated with 20 mM Hepes pH 7.5, 150 mM NaCl, 5 mM MgCl₂, 5% glycerol and 1 mM DTT (buffer A). For data collection in the presence of nucleotides, the nucleotide-free proteins were pre-incubated with 1 mM of the nucleotide, and 200 μM ( = molar excess) of the nucleotide was added to buffer A. Initial processing of the data was done with DATASW[46]. PRIMUS was used for determination of the radius of gyration (Rg) using the Guinier approximation[47], and GNOM for the calculation of the pair distance distribution function P(r)[48]. Modelling of missing loops in the crystallographic dimer model of the CtRoc-COR was done using ModLoop[49] and the missing N- and C-terminus were added with CORAL[50]. The final dimer model was compared with the experimental data with CRYSOL[51]. *Ab initio* envelopes were calculated using DAMMIN (average of 19 runs for CtRoc-COR and 20 runs for CtRoco)[52] followed by DAMAVER[53]. The final model was generated using one round of DAMMIN[52], starting from the damstart model generated by DAMAVER. Docking into the envelope was performed with Supcomb[54]. For docking into the CtRoco envelope, the position of the CtRoc-COR dimer structure was manually adjusted. The molmap command in chimera was used to convert the *ab initio* bead models into 20 Å density maps[55]. An overview of the experimental and modelling parameters is provided in Supplementary Tables 1–3.

For the SEC-MALS experiments, the samples were prepared and run in the same way as for the SEC-SAXS experiments. 10 μl of an 8 mg ml⁻¹ protein sample was injected on a Bio SEC-3 HPLC column (Agilent, 3 μm 300 Å). A Dawn Heleos detector (using 9 angles) and Optilab T-rEX detector (Wyatt technology) were attached to a HPLC (Shimadzu). The molar masses were calculated with the ASTRA 5.3.4.20 software.

**Native mass spectrometry.** Protein samples were exchanged to a buffer containing 150 mM ammonium acetate pH 7.5 using micro Bio-spin columns (Bio-gel P6, Bio-rad). For measurements in the presence of nucleotide (GDP or GppNHp), samples were pre-incubated with 500 μM nucleotide and 100 μM nucleotide was added to the ammonium acetate buffer. Final protein concentrations ranged between 12.5 and 16 μM. Samples were introduced into the vacuum of the mass spectrometer using nanoelectrospray ionization with in-house-prepared, gold-coated borosilicate glass capillaries with a spray voltage of +1.4 kV. Spectra were recorded on a quadrupole TOF instrument (Q-TOF2, Waters) modified for transmission of native, high-m/z protein assemblies, as described elsewhere[56]. Critical voltages and pressures throughout the instrument were 120 and 25 V for the sampling cone and collision voltage respectively, with pressures of 10 and 2E−2 mbar for the source and collision cell.

**Analytical ultracentrifugation.** Sedimentation velocity (SV-AUC) experiments on CtRoco and CtRoc-COR were carried out at 20 °C in 20 mM HEPES pH 7.5, 150 mM NaCl, 5 mM MgCl₂, 5% Glycerol and 100 μM of the respective nucleotides on a Beckman Coulter ProteomLab™ XL-I analytical ultracentrifuge using the absorbance at 280 nm. Samples were prepared at a concentration of 0.57 mg ml⁻¹ (4.5 μM) for CtRoco and 0.3 mg ml⁻¹ (4.6 μM) for CtRoc-COR. Standard double sector centrepieces were used. The cells were scanned every minute and in total 200 scans were collected. The data was analysed using SEDFIT 15.01b[57] with the continuous c(s) distribution model. Solution density $\rho$, viscosity $\eta$ and partial specific volumes $\bar{v}$ were calculated using SEDNTERP[58] ($\rho$ = 1.02061 g l⁻¹; $\eta$ = 0.01197 kg s⁻¹ m⁻¹, $\bar{v}$ (CtRoco) = 0.7446, $\bar{v}$ (CtRoc-COR) = 0.7398). The c(s) analysis was carried out with an s range of 0 to 15 with a resolution of 200 and a confidence level of 0.68. In all cases, fits were good, with root mean square deviation (rmsd) values ranging from 0.005 to 0.012. Results were prepared for publication using GUSSI 1.2.1[59].

Sedimentation equilibrium experiments (SE-AUC) on CtRoco were conducted on the same instrument and under the same buffer conditions as SV-AUC experiments using an Epon six-channel centrepiece and measurement of the absorbance at 280 nm. Each sample was used at three different concentrations (0.9 mg ml⁻¹/7.2 μM, 0.58 mg ml⁻¹/4.5 μM, and 0.35 mg ml⁻¹/2.7 μM). Samples were centrifuged at 1164, 3407 and 9757 xg until sedimentation equilibrium was reached. Three scans were taken at 280 nm at each point. The data was analysed with a SEDPHAT[60] software using either monomer-dimer or single-species model. Figures were generated using GUSSI 1.2.1[59].

**FRET and fluorescence stopped flow.** The S928C mutant was generated in a cysteine-free variant of CtRoc-COR using quick-change mutagenesis, as described earlier[19]. 10 mg of completely reduced pure CtRoc-COR S928C protein was loaded on a Superdex S200 16/60 column (GE Healthcare) equilibrated with a degassed buffer composed of 20 mM Hepes pH 7, 150 mM NaCl, 5 mM MgCl₂ and 5%

glycerol. Peak fractions were used to perform the labelling reaction with 8 μM protein, 36 μM maleimide-Cy3 (Lumiprobe) and 36 μM maleimide-Cy5 (Lumiprobe) in a total volume of 5 ml. After incubation at room temperature for 2 h, the unreacted fluorophores were separated from the labelled protein using size-exclusion chromatography on a S200 16/60 column equilibrated with 20 mM Hepes pH 7.5, 150 mM NaCl, 5 mM MgCl₂, 5% glycerol and 1 mM DTT. Labelling stoichiometry was spectrophotometrically determined to be approximately 30% for each fluorophore. To determine the monomerization rate, 0.2 μM double labelled protein was mixed with 100 μM nucleotide in a stopped flow apparatus (Applied Photophysics). To follow monomerization during single GTP-turnover, 2 μM Cy3/Cy5-labelled CtRoc-COR was rapidly mixed with 2 μM GTP. The Cy3 fluorophore was excited at 540 nm and change in Cy5 emission was monitored using a cut-off filter of 645 nm.

Mant-GTP binding and hydrolysis was followed in a stopped-flow apparatus (Applied Photophysics) by rapidly mixing 2 μM of protein with 2 μM of mant-GTP. The mant-fluorophore was excited at 360 nm and emission was followed through a cut-off filter of 405 nm. All experiments were performed at 25 °C and at least 3 time traces were averaged.

**Negative stain electron microscopy.** For negative stain electron microscopy, 2 μL of a 0.01 mg ml⁻¹ protein sample was applied on a glow discharged carbon-coated copper grid. After three short wash and blot steps with MilliQ water, the grids were stained with a 1% uranyl formate solution. Grids were visualized with a JEOL JEM-1400 electron microscope operating at 120 kV and equipped with a LaB₆ cathode. Images were recorded on a CMOS TemCam-F416 camera (TVIPS, Germany) at a nominal magnification of 80,000, a defocus of approximately 2 μm and a corresponding pixel size of 1.42 Å.

For imaging the different nucleotide-bound states, 0.01 mg ml⁻¹ CtRoco protein solutions were pre-incubated with 1 mM GDP or GppNHp. In the multiple turnover experiment, 1 mM GTP/GppNHp was added to 0.01 mg ml⁻¹ CtRoco protein. For the single turnover experiment, where turnover of 1 μM GTP by 1 μM CtRoco was followed, samples were diluted to 0.01 mg ml⁻¹ right before spotting. In both experiments samples were taken at the following time points: 0, 2, 4, 6, 25, 50 and 75 min. The time-resolved EM experiments were performed at 25 °C. For the calculation of class averages, 11,571 particles for the nucleotide-free state, 9620 particles for the GDP-bound state and 11,164 for the GppNHp-bound state were selected using e2boxer[61]. Further classification was done with SPARX[62].

**GTP hydrolysis assays.** GTP hydrolysis experiments were performed at 25 °C in 20 mM Hepes pH 7.5, 150 mM NaCl, 5 mM MgCl₂, 5% glycerol and 1 mM DTT. The GTP-GDP mixture was separated using a C18-reversed phase column (Phenomenex, Jupiter 5 μm C18 300 Å) coupled to a HPLC system (Waters), using 100 mM KH₂PO₄ pH 6.4, 10 mM tetra butyl-ammonium-bromide and 7.5% acetonitrile as a mobile phase. Nucleotide elution was followed using absorbance at 254 nm. The area of GDP was converted to concentration using a standard curve.

Time points for the single turnover experiments with 1 μM protein and 1 μM GTP were 0, 2, 6, 13, 20, 30, 40, 60, 80 and 100 min. For the multiple turnover steady state experiments 0.2 μM protein was incubated with different concentrations of GTP (25, 50, 100, 175, 250 μM) and samples were taken after 0, 30, 60, 90 and 120 min. Kinetic parameters were determined by fitting the data to a single exponential (single turnover) or the Michaelis-Menten equation (multiple turnover) using GraphPad Prism 6. All experiments were done in triplicate.

**Data availability.** All data supporting the findings of this study are available from the corresponding author upon reasonable request. All SAXS data and derived models were deposited in the SASDB (CtRocCOR NF: SASDCB2, CtRocCOR GDP: SASDCC2, CtRocCOR GppNHp: SASDCD2, CtRoco NF: SASDC82, CtRoco GDP: SASDCA2, CtRoco GppNHp: SASDC92, CtRoco L487A NF: SASDCG2, CtRoco L487A GDP: SASDCE2, CtRoco L487A GppNHp: SASDCF2).

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

## Acknowledgements

We would like to thank the staff at the beamlines SWING of Soleil (France) and BM29 of ESRF (France) for assistance during data collection. This work was supported by the Fonds voor Wetenschappelijk Onderzoek (M.L., R.E., W.V.), BioStruct-X by the European Community's Seventh Framework Programme (W.V.), a Strategic Research Program Financing of the VUB (W.V.), VUB/RUG collaboration agreement (OZR2544; L.W.), the Hercules foundation (W.V.), The Michael J. Fox Foundation for Parkinson's Research (P.V., Ar.K., G.G., C.J.G., W.V.) and a NWO-VIDI grant (Ar.K.). We thank Annelore Stroobants for assistance with electron microscopy, Pragya Pathak for help with data analysis and Roise Mc Govern for carefully reading the manuscript.

## Author contributions

E.D., L.W., P.V., A.P., R.E., C.J.G., F.S., Ar.K. and W.V. designed the experiments. E.D., G.G., M.L., H.P., L.S., P.A. and S.T. purified the proteins. L.W. and R.E. performed E.M. analysis. E.D. and R.G. performed the SEC-MALS analysis. E.D. and W.V. performed the SEC-SAXS analysis. S.T., A.P. and E.D. performed the AUC analysis. Al.K. and E.D. performed the native MS analysis. E.D. performed the kinetic and FRET experiments. E.D. and W.V. wrote the manuscript. All the authors read and edited the manuscript.

## Additional information

**Competing interests:** The authors declare no competing financial interests.

