## [Peer Review file · Nature Communications]

Reviewers' comments:

Reviewer #1 (Remarks to the Author):

The manuscript by Deyaert et al. shows that a construct of *C. tepidum* Roco protein appears to undergo a conformational change from dimer to monomer upon GTP binding. This is potentially interesting because the tandem Roc-Cor structure is also found in a Parkinson's disease-associated protein called LRRK2, thus insights gleaned from the *C. tepidum* Roc-Cor (CtRoc-COR) construct might also shed light on LRRK2. While the functional significance of the reversible monomer-dimer transition is unclear, it is nonetheless interesting.

The data convincingly show that CtRoc-COR undergoes monomer-dimer conformational changes upon GTP and GDP binding. However, I have a few reservations.

Major issues:

1. The EM images of the dimeric structure of *C. tepidum* Roco look different from the published EM images of LRRK2 (Guitoli et al. PNAS 2016), thus it is unclear whether or not the conformational changes observed for CtRoco could be inferred for LRRK2 activity.
2. The EM images of the dimeric structure of CtRoc-Cor presented in figures 1g do not match the SAXS model presented in figure 1b, which in turn do not match the crystal structure presented in figure 2a. Thus it is unclear which conformations or shapes represent the biological structures.

Minor issues:

1. Figure 1f; it is unusual that the 173 kDa peak elutes at the same volume as the 252 kDa peak.
2. Figure 1b; the cartoon ribbon model does not fill the SAXS envelope.
3. Figure 2b; the range of the normalized fluorescence change is narrow (from 0.85 to 1.0), please show raw AU readings.
4. Figure 2b legend; specify the concentration of GppNHp.
5. Page 10 to 11; here the authors show that the dimeric conformation of CtRoco has decreased GTPase activity, which contradicts with the literature's suggestion that dimeric formation is necessary for GTPase activity (Gotthardt et al. EMBO J. 2008).

Reviewer #2 (Remarks to the Author):

The manuscript "A bacterial homologue of the Parkinson's disease-associated protein LRRK2 undergoes a monomer-dimer transition during GTP turnover" by the Versees group reports on the biophysical characterization of the Roco protein from the bacterium *Chlorobium tepidum*. Structural and biophysical characterizations of Roco proteins may have some significance since they could open a window into the understanding of the mammalian LRRK2 protein. Mutations of LRRK2 are implicated in hereditary forms of Parkinson disease (PD). Therefore, analyzing the functions and biochemical activities of LRRK2 is important for the understanding of PD. However, LRRK2 is notoriously challenging to study on the biochemical and structural level since its expression and purification only gives small protein amounts. Consequently, bacterial homologues such as Roco have been used as a surrogate for studying LRRK2.

Studies conducted by the Versees group now characterize the homodimerization of Roco in dependence of binding to guanine nucleotides (i.e. GTP, GDP, GppNHp). Since Roco contains a nucleotide binding domain with GTP-hydrolyzing activity, the researchers investigated the influence of nucleotide-binding and -hydrolysis on the formation of homodimeric complexes. Using state-of-the-art methods, the lab could nicely demonstrate the switching of dimer and monomer states upon GDP or GTP binding. They also could show that the dimerization operates on a similar time scale with GTP-hydrolysis, demonstrating that GTP can actively dissociate the Roco

homodimer under physiological conditions. Furthermore, when they introduce a PD-analogous LRRK2-mutation in Roco, the dimerization and nucleotide hydrolysis is affected, potentially pointing at a molecular mechanism operating in PD, too.

The experiments are carried out with great diligence. The conclusions of the authors are fully justified on the basis of their data. Technically, the manuscript is excellent and the data are very well presented.

Nevertheless, I doubt that the level of insight obtained by the authors justifies a publication in Nature Communications. The authors present a functional characterization of a bacterial protein but claim that this potentially provides insights into the activities of the mammalian PD-associated LRRK2 protein. This is probably a bit too far-fetched and too artificial. Also, there are several structures available for Roco and related proteins and their dimerization properties have also been studied. Consequently, the manuscript is suited better for a more specialized journal. However, if the authors could demonstrate with experiments that the same dimerization properties apply also to LRRK2, this may be a different story.

Minor comments:

- The presentation of the EM-data require optimization. The negative stain images are badly recognizable and a discrimination of different dimerization states on their basis is hardly possible for the unfamiliar reader.

Reviewer #3 (Remarks to the Author):

The main result extracted from the SAXS data is that both CtRoc-COR and CtRoco are dimeric in native form and upon addition of GDP and become essentially monomeric upon addition of GppNHp. The results are corroborated by light scattering and negative stain EM, and further rationalized in terms of the consideration of the GTPase cycle.

These results are undoubtedly interesting and would have been worth publishing. The problem is however that, as presented, the experimental SAXS data do not appear to fully agree well with the conclusions of the authors. Appropriate revision of the Ms is therefore required.

Indeed, Figure 1a,e depict the six relevant X-ray scattering curves displaced in logarithmic scale.

Looking at Figure 1a, the CtRoc-COR GDP curve (green) is admittedly essentially parallel to the NF indicating that addition of GDP does not change the overall shape. However, the discrepancy of the fit for CtRoc-COR GDP ($\chi=6.4$) is very large not only compared to the NF ($\chi=1.1$) but also compared to CtRoc-COR GppNHp ($\chi=3.3$), which is not understandable, as the latter is supposed to be a monomer.

Overall, do the authors report χ or χ^2 values?

For the CtRoco, the situation is yet more worrisome. Contrary to what the authors write (lines 113-115),

in Fig. 1e, the green and red curves (corresponding to GDP and GppNHp bound states) are essentially parallel to each other and they both noticeably differ from the black curve (NF). From this plot it follows that the GDP and GppNHp states are similar to each other and differ from NF. Further, in Supplementary Table 2, the radius of gyration of CtRoco GppNHp (6.5 nm) is close to that of CtRoco GDP (6.2 nm) and both significantly exceed the Rg of CtRoco MF (5 nm). The authors must explain this controversy.

The authors present ab initio shape for CtRoc-COR just to show that it overlaps with the crystal structure. Why do they not provide the shapes of CtRoco, which would have been much more interesting?

In lines 119-120 the authors substantiate the presence of dimers by "relatively high protein concentration used in these experiments (8 mg/ml)". This is actually the initial solute concentration -- is the protein not substantially diluted on the column?

The motivation and the gain for the authors to conduct in-line SEC/SAXS experimnt is not clear. These experimnts, which are much longer and more complicated than standard SAXS measurements, are conducted if the samples are notoriously polydisperse and cannot be purified to have clean solutes for the batch analysis. In the SEC data displayed in the Ms, single peaks are shown, and it was not indicated what was the advantage of SEC/SAXS in this case. Moreover, for CtRoco GDP/GppNHp samples, the authors speak about dimer/monomer mixtures, (where SEC/SAXS should indeed be useful) -- why are these mixtures not separated at all on the column?

Minor comments:

It appears that there are some mistakes in Refs 42, 52, 54.

Numerous times in the text and supplementary, the authors write "scatter curves" -- they should use "scattering curves".

Finally, SAXS data/models should be properly deposited (www.sasbdb.org) and the accession codes reported in the publication.

Reviewer #4 (Remarks to the Author):

In their manuscript, Deyaert et al. use SAXS, MALS, FRET and EM to describe Roco upon treatment with nucleotides and their analogue in comparison to the apo-form. Based on their data, the authors suggest a model for a monomer-dimer transition of the bacterial Roco protein dependent on the nucleotide state. The data presented are, however, not sufficient to differentiate between the suggested monomer-dimer transition and a conformational rearrangement. Furthermore, the EM images shown are of poor quality in particular upon nucleotide treatment.

Major points:

1. Upon addition of GDP, GppNHp, or GTP, Roco/CtRoco aggregates (e.g. Fig. 1g; 2c, 3c). The GppNHp state appears to adopt a less compact, elongated, "worm"-like shape. These samples do not appear to be just "half" of the nucleotide state. Along these lines, the authors also state that "the radii of gyration ... do not change much upon nucleotide binding suggesting that the protein

subunits become elongated in their monomeric form". Furthermore, the authors admit the "occurrence of secondary inter-domain conformational changes in the CtRoco monomers compared to the dimers". The data presented are, however, not sufficient to differentiate between the suggested monomer-dimer transition and a conformational rearrangement and further experiments are needed to clarify these points. In order to provide further support for a dimer-monomer transition suggested by the authors, 3D structures calculated from the SAXS and EM data as well as immuno-electron microscopy (using antibodies that allow differentiation between the monomeric and the dimeric state) coupled to statistical analysis are required.

2. Why did the particles aggregate upon treatment (e.g. Fig. 1g, middle and right, upper panel; Fig. 2c and 3c)? The strong aggregation makes it challenging to judge the shape of the particles, and accordingly, the authors also did not provide any class averages for GDP state and the time-courses they present. Class averages for all samples are required.

3. Fig. 1g, GppNHp: How do the authors explain the "extra" densities in the class averages? How do the authors exclude that these are rearranged particles?

4. Fig. 3c "CtRoco is almost completely converted from a dimeric to a monomeric form after 240 seconds." The EM images do not support this claim. Instead, severe aggregation is observed.

5. L. 82: Ab initio models based on the SAXS data should be provided for all data sets as listed in Suppl. Table 1-3.

6. Models need to be deposited in a relevant database such as <https://www.sasbdb.org/> and accession numbers indicated in the manuscript.

7. L. 82: Delete "very good agreement" and provide additional data. The fit looks rather random. How was the crystal structure fitted into the ab initio model? Are there alternative fits and which support do the authors have to exclude the alternative fits?

8. L. 343ff: The description of the EM methods does not provide sufficient information to allow the reader to judge whether or not the data are sound. How did the authors exclude that the treatment (e.g. washes and drying steps) influence the shape of the particles?

9. L. 417, ref. 21: The authors should discuss this reference in detail. A dimeric state of LRRK2 was found in the presence of GppNHp in ref. 21 – how do the authors explain these findings? How does the data presented in ref. 21 fit to the authors' model? The authors should also prepare a figure, where they compare their model and the previously published model (ref. 21).

Minor points:

10. L. 31-32: Delete "thereby potentially opening new avenues towards drug design". The manuscript does not provide data on drug design.

11. L. 234: Delete "for the first time unequivocally".

12. L. 290: Please provide more details for "lung and kidney abnormalities".

13. L. 291ff: Delete this sentence as the manuscript does not provide sufficient support.

14. L.354: How many particles were averaged per class in average?

15. L. 399: A space is missing in "thatstimulates".

Manuscript No.: NCOMMS-16-22070

A bacterial homologue of the Parkinson's disease-associated protein LRRK2 undergoes a monomer-dimer transition during GTP turnover

Point-by-point response to referee remarks

We would like to thank all referees for their thorough revision of our manuscript, their positive comments and their useful and constructive remarks.

Below is a point-by-point response to all the referee remarks and a discussion of several additional experiments that have been included in our revised manuscripts to further support our conclusions.

Reviewer 1:

The manuscript by Deyaert et al. shows that a construct of C. tepidum Roco protein appears to undergo a conformational change from dimer to monomer upon GTP binding. This is potentially interesting because the tandem Roc-Cor structure is also found in a Parkinson's disease-associated protein called LRRK2, thus insights gleaned from the C. tepidum Roc-Cor (CtRoc-COR) construct might also shed light on LRRK2. While the functional significance of the reversible monomer-dimer transition is unclear, it is nonetheless interesting. The data convincingly show that CtRoc-COR undergoes monomer-dimer conformational changes upon GTP and GDP binding. However, I have a few reservations.

We thank the referee for the positive comments.

Major Remarks

1. The EM images of the dimeric structure of C. tepidum Roco look different from the published EM images of LRRK2 (Guaitoli et al. PNAS 2016), thus it is unclear whether or not the conformational changes observed for CtRoco could be inferred for LRRK2 activity.

The negative stain EM images of CtRoco shown in this manuscript and the previously published LRRK2 (Guaitoli *et al.*, PNAS 2016) indeed look different. However, we would like to clarify that it is expected that they differ significantly. The CtRoco protein used here for EM consists of an LRR domain, the Roc-COR tandem domain and a C-terminal domain of unknown structure. LRRK2 is much bigger and contains, apart from the LRR and Roc-COR domains also several other domains (Armadillo, Ankyrin, Kinase, WD40).

The structural model of LRRK2 published in Guaitoli *et al.*, PNAS 2016 was based on a combination of homology modelling of the individual domains, distance restriction based on cross-linking experiments and negative stain EM. To build the LRRK2 model the LRR-Roc-COR domain arrangement was used as a starting point around which all the other domains were placed. This LRR-Roc-COR model was itself based on the available structures of the LRR domain (5IL7) and the Roc-COR domain (3DPU) of the *C. tepidum* Roco protein, by superimposing a common helix present in the structure of both the LRR domain and Roc-COR domain. Hence, the very same structures of the CtLRR and CtRoc-COR that we study here were also used to build the LRRK2 model.

To validate the EM images that we report in our manuscript in a very qualitative way we can take the model of CtRoco generated by superimposing the CtLRR and CtRoc-COR structures via the common α -helix, and use this to generate theoretical 2D EM projections using the program EMAN2 (see Reviewer Fig1 a & b). As is shown in Reviewer Figure 1c, several of these theoretical EM images

correspond very well to our experimentally observed EM class averages of the nucleotide free dimeric CtRoco (note however that in CtRoco used in EM an additional C-terminal domain of 150 amino acids is present that is not present in the structure model). This observation is a very good indication that the negative stain EM images that we show in our manuscript are a good representation of CtRoco and of the LRR-Roc-COR domain arrangement that was used in generating the model of LRRK2 in Guaitoli *et al.*, PNAS 2016.

Reviewer Fig. 1: Comparison of theoretical 2D EM projections of a LRR-Roc-COR model of CtRoco with experimental negative stain EM class averages of CtRoco. (a) Model of dimeric CtLRR-Roc-COR based on alignment of a common helix of the CtLRR (yellow) and CtRoc-COR (Roc domains in red, COR domains in blue) structures. (b) Theoretical 2D EM projections of dimeric CtLRR-Roc-COR. (c) Alignment of 5 theoretical 2D EM projections (top) with 5 experimental class averages of nucleotide free CtRoco (bottom) showing that the observed particles in negative stain EM are in good agreement with the expected shape of CtLRR-Roc-COR EM images.

Because the Roc-COR module is conserved among all Roco proteins and considering that it takes a central role in LRRK2 (see above and below), we do firmly believe that our results are also of high relevance for LRRK2 functioning. It has indeed already been shown that also in LRRK2 the COR domain plays an important role in dimerization (e.g. Terheyden *et al.*, Biochem. J. 2014). Of course we can at this point not exclude that other domains of LRRK2, not present in the bacterial homologue also play a role in oligomerisation and that the mechanism might even be more complex in the human protein. The latter aspects are now more explicitly discussed in a dedicated paragraph in the Discussion section.

2. The EM images of the dimeric structure of CtRoc-Cor presented in figures 1g do not match the SAXS model presented in figure 1b, which in turn do not match the crystal structure presented in figure 2a. Thus it is unclear which conformations or shapes represent the biological structures.

We would like to clarify that two different constructs have been used in our study: (1) a construct consisting only of the Roc and COR domains (called CtRoc-COR in the paper), which is shown in Fig. 1b and 2a (using original figure numbering) and (2) the full length protein (called CtRoco in the paper), consisting of a LRR-Roc-COR-C-terminal domain, which is used for EM in Fig. 1g. We start

the paper by using the CtRoc-COR construct since this is the only protein for which a crystal structure is available (this is in fact the only structure available so far of any Roc-COR domain). Hence, this protein allows a straightforward interpretation of e.g. SAXS data in comparison with the crystal structure. However, the size of this domain construct is on the borderline for EM experiments (130 kDa as a dimer). Subsequently we use the full length protein (CtRoco), which is also used in the EM studies. However, all conclusions concerning the monomer/dimer equilibrium are valid for both variants.

To clarify the use of these two different constructs in the manuscript we have split up Figure 1 in two figures, one concerning CtRoc-COR (Fig. 1) and one concerning CtRoco (Fig. 2).

Minor remarks

1. *Figure 1f; it is unusual that the 173 kDa peak elutes at the same volume as the 252 kDa peak.*

(Note that figure 1f is figure 2d in the revised manuscript.)

This seems indeed remarkable at first sight. However, separation of macromolecules via size exclusion chromatography is based on their Stokes radius and not on molecular weight. Consequently, an elongated protein can elute at the same volume as a globular protein with twice its molecular weight (see for example Harold P. Erickson, ‘Size and Shape of Protein Molecules at the Nanometer Level Determined by Sedimentation, Gel Filtration, and Electron Microscopy’, *Biological Procedures Online*, 11.1 (2009), 32–51). We believe that it is this what we observe for the CtRoco protein. We hypothesize that GppNHp-induced monomerization is accompanied by an elongation of the individual subunits, which results in roughly the same elution volume of the monomeric and dimeric proteins. Clear indications for such an elongation are provided by SAXS analysis that shows that the R_g and D_{max} values of GppNHp-bound CtRoco are larger than the values of nucleotide-free CtRoco. Also EM shows more elongated “worm like” particles for GppNHp-bound CtRoco as compared to the X-shaped particles for the nucleotide-free protein.

MALS, which is shape-independent, does however clearly show the decrease in MW in the GppNHp state. Moreover, we also included in the revised document native MS, sedimentation velocity analytical ultracentrifugation and sedimentation equilibrium analytical ultracentrifugation experiments that all show the same shift toward the monomeric state in the presence of GppNHp (see new Fig. 1 and Fig. 2).

We also discuss the observation that both protein forms elute at the same position on SEC more explicitly in the text (p 7-8): “It might seem remarkable at first sight that despite the difference in molecular mass the nucleotide-free, GppNHp- and GDP-bound CtRoco proteins elute at approximately the same volume in SEC (Fig. 2d). Correspondingly, Guinier analysis of the SAXS data shows that the radius of gyration (R_g) even increases upon going from the nucleotide-free state, over the GDP-bound state to the GppNHp-bound state (Supplementary Fig. 4). This could be explained by a conformational change occurring in the CtRoco subunits upon monomerization or upon nucleotide binding. Indeed, as is the case for the CtRoc-COR construct, the SAXS pair-distance distribution function of CtRoco shows a shift toward shorter average inter-atom distances in the presence of GppNHp compared to GDP or nucleotide-free states, indicative of monomerization (Supplementary Fig. 4d). However, the curve of the GppNHp-bound CtRoco protein also shows a tailing toward longer maximal distances. This strongly suggests the occurrence of a GppNHp-bound monomer that is more elongated than the corresponding subunits in the nucleotide-free dimer. Also in the GDP-bound state tailing of the pair-distance distribution function is observed indicating that already some conformational changes occur in the dimeric nucleotide-bound protein. These conformational changes also explain why the observed monomer/dimer equilibrium would probably be missed when assessed only with size-exclusion chromatography.”

2. Figure 1b; the cartoon ribbon model does not fill the SAXS envelope.

We do not agree with the referee that the SAXS envelope does not fit with the crystal structure. Overall the fit is very good and comparable to other envelopes generated from SAXS (e.g. Nat Commun. 2015 Jun 10;6:7380. doi: 10.1038/ncomms8380). We would like to emphasize that SAXS is a low resolution method.

Moreover, in light of a remark made by Reviewer 3 we have re-measured all the SAXS data of CtRoc-COR on the SWING beamline to have a consistent X-ray source for all nucleotide states. Using these data we also recalculated the *ab initio* envelope which fits the structure even better (see new Fig. 1b).

3. Figure 2b; the range of the normalized fluorescence change is narrow (from 0.85 to 1.0), please show raw AU readings.

The non-normalised data are shown in Reviewer Fig. 2a. Since the initial fluorescence reading for the GppNHp trace is somewhat higher than the other readings, we prefer to use the normalized data for clarity. This however does not take away any of the conclusions we draw from these data, since we are interested in the time-dependency of the signal, and we compare the data of GppNHp and GTP with GDP and buffer controls.

The reason for the seemingly narrow range of decrease in FRET signal is probably the high fluorescence background. A first source of background is the FRET signal of a random Cy3/Cy5 mixture (i.e. the FRET signal after monomerization). More importantly, we measure the FRET signal with a 645 nm cut-off filter. As you can see in Reviewer Fig. 2b, a significant background of Cy3 emission is expected above 645 nm, resulting in a total signal that is the sum of a FRET signal and a direct Cy3 emission.

Reviewer Fig. 2: Stopped flow FRET experiment using labeled CtRoc-COR (a) Non-normalized stopped-flow FRET traces of Cy3/Cy5-labeled CtRoc-COR (S928C) upon mixing with buffer (black), GDP (green), GppNHp (red) or GTP (blue). (b) Excitation and emission spectra of Cy3 (green) and Cy5 (red). Overlap of Cy3 emission and Cy5 excitation, giving rise to FRET, is coloured blue. The wavelength cut-off of the filter (645 nm) used in the FRET experiment is indicated with a black dashed line. Cy3 shows a significant fluorescence emission above 645 nm (indicated by the black dashed rectangle) that will contribute to the fluorescence background of our experiment. (Figure adapted from Ishikawa-Ankerhold et al. *Molecules* 2012).

4. Figure 2b legend; specify the concentration of GppNHp.

The figure legend (Fig. 3b in revised document) now reads “The FRET signal of Cy3/Cy5-labeled CtRoc-COR (S928C) over time is shown after rapid mixing with buffer (black), or 50 μ M GDP (green), GppNHp (red) or GTP (blue).”

A consistent concentration of 50 μ M for all nucleotides is used.

5. Page 10 to 11; here the authors show that the dimeric conformation of CtRoco has decreased GTPase activity, which contradicts with the literature's suggestion that dimeric formation is necessary for GTPase activity (Gotthardt et al. EMBO J. 2008).

In Gotthardt *et al.* a construct was used that lacks the entire C-terminal half of the COR domain rendering it constitutively monomeric (Roc-COR Δ C). This protein was indeed reported to have decreased GTPase activity. This decrease in GTPase activity might however be due to the deletion of a significant part of a domain per se. Alternatively, the decrease in GTPase activity might also be caused by a permanent monomerization of the protein. In fact our model, in which the Roco proteins need to cycle between a monomeric and dimeric form, would predict a reduced activity both for a permanent monomer and a permanent dimer.

This is now explicitly mentioned in the discussion (p 15): “According to our model, whereby the CtRoco protein needs to cycle through a monomeric and dimeric state, stabilization of either could lead to a reduced GTPase activity.”

Reviewer 2:

*The manuscript “A bacterial homologue of the Parkinson’s disease-associated protein LRRK2 undergoes a monomer-dimer transition during GTP turnover” by the Versees group reports on the biophysical characterization of the Roco protein from the bacterium *Chlorobium tepidum*. Structural and biophysical characterizations of Roco proteins may have some significance since they could open a window into the understanding of the mammalian LRRK2 protein. Mutations of LRRK2 are implicated in hereditary forms of Parkinson disease (PD). Therefore, analyzing the functions and biochemical activities of LRRK2 is important for the understanding of PD. However, LRRK2 is notoriously challenging to study on the biochemical and structural level since its expression and purification only gives small protein amounts. Consequently, bacterial homologues such as Roco have been used as a surrogate for studying LRRK2.*

The experiments are carried out with great diligence. The conclusions of the authors are fully justified on the basis of their data. Technically, the manuscript is excellent and the data are very well presented.

We thank the referee for this positive comment.

*Nevertheless, I doubt that the level of insight obtained by the authors justifies a publication in *Nature Communications*. The authors present a functional characterization of a bacterial protein but claim that this potentially provides insights into the activities of the mammalian PD-associated LRRK2 protein. This is probably a bit too far-fetched and too artificial. Also, there are several structures available for Roco and related proteins and their dimerization properties have also been studied. Consequently, the manuscript is suited better for a more specialized journal. However, if the authors could demonstrate with experiments that the same dimerization properties apply also to LRRK2, this may be a different story.*

We do not agree with the referee on these last points. The presence of the Roc-COR supradomain is highly conserved among all Roco proteins, including LRRK2, and is also the namesake of this protein family. As such it can be expected that the main function of this domain is widely conserved. We would moreover like to stress that so far only one structure is available of a complete Roc-COR module, being the one from *C. tepidum*, which is also used in our study. The two other structures available in the pdb either represent only the Roc domain (Deng *et al.*, PNAS 2009) or a Roc-COR deletion construct from *M. barkeri* where the C-terminal half of the COR domain has been deleted (Terheyden *et al.*, Biochem J 2014). As such the crystal structure of the dimeric nucleotide-free Roc-COR from *C. tepidum* has played a key role in the currently widely accepted working models for the functioning of the Roc-COR domain of LRRK2, and LRRK2 as a whole. One of the main working hypothesis based on the crystal structure from CtRoc-COR is that Roco proteins, including LRRK2, belong to the G proteins activated by nucleotide dependent dimerization (GAD). This model implies that COR acts as a permanent dimerization device and that the Roc domains complement each other upon GTP binding. This has been discussed in several reviews, to name just a few: Gasper *et al.*, Nat. Rev. Mol Cell Biol. 2009; Nixon-Abell *et al.*, Biochem. Soc. Trans. 2016; Gilsbach & Kortholt, Front Mol Neurosci. 2014.

Our results now show that the classification of Roco proteins as classical GADs needs reconsideration. We show that the CtRoco protein is not a permanent dimer, and that it moreover undergoes a dimer-monomer cycle concomitant with GTP binding and hydrolysis. This means that the model for the functioning of the Roc-COR module is much more complex than anticipated, which also has clear consequences for our thinking about LRRK2.

Moreover, similar to the prokaryotic Roco proteins, various studies report that LRRK2 can form dimers through its COR domains *in vitro* (e.g. Terheyden *et al.* Biochem. J. 2014 and references given therein). However, other studies suggest that LRRK2 is a monomer or exists as a monomer dimer

equilibrium *in vivo* (Nixon-Abell *et al.* Front. Mol. Neurosci, 2016; Ito & Iwatsubo Biochem J. 2012; James *et al.* Biophys J 2012, ...). As now discussed in more detail in the revised manuscript, evidence is accumulating that the oligomeric state of LRRK2 is dependent on its localization in the cell. Experiments with cell fractionation have shown that dimeric LRRK2 was present in the membrane fraction, while the majority of LRRK2 was monomeric and present in the cytosolic fraction (Berger *et al.*, 2010). This LRRK2 monomer/dimer equilibrium was confirmed in living CHO-K1 cells by James *et al.* (James *et al.*, 2012). In these experiments, LRRK2 was fluorescently labeled with GFP and oligomerization was measured by confocal microscopy in combination with Number and Brightness analysis. *[Redacted]*

[Figure redacted]

These findings agree with our results on CtRoco and can now be re-evaluated in light of the results presented in our paper. Of course, we are fully aware that apart from the LRR, Roc and COR domains, LRRK2 contains several other domains potentially contributing to the dimer interface and which could make the mechanism even more complicated in LRRK2. In addition regulatory factors, such as membrane binding, phosphorylation or interaction with other proteins, most likely influence the monomer-dimer equilibrium of LRRK2 *in vivo*. In order to avoid the risk that the reader overstretches our results in a linear way to LRRK2, and in order to better emphasize the importance and caveats of our results in the light of LRRK2 functioning, we have significantly re-written the introduction part and discussion part of our paper to address these issues.

Notwithstanding the points made above we do agree with the referee that it would be even better if we could additionally prove in a quantitative way the nucleotide-induced monomerization of LRRK2 *in vitro*. However, as the referee points out LRRK2 is a notoriously challenging protein. The protein is expressed in HEK293T cells and yields are extremely low. The protein quickly degrades after purification such that it needs to be used at the same day of purification and the protein has a severe tendency to aggregate and stick to a size exclusion chromatography resin (also hampering SAXS and MALS) (see Reviewer Fig. 4). Moreover, the protein is not stable in a nucleotide free state such that a homogeneous loading with a particular nucleotide cannot be assured.

Reviewer Fig. 4: Purification of LRRK2 from HEK cells. Strep/Flag (SF)-tagged LRRK2 was expressed in HEK293T cells and purified on Strep-Tactin Superflow resin. **(a)** Representative SDS-PAGE with lane 1: Spectra Multicolor High Range Protein Ladder, lane 2: Total lysate of HEK293T cells and lane 3: LRRK2 after purification. **(b)** Western blot, using Anti-LRRK2 antibody MJFF2 (c41-2), of LRRK2 one day after purification. The Western blot shows severe degradation of LRRK2. In-gel digestion mass spectrometry analysis confirmed C-terminal degradation.

Despite these serious limitations we have intensively tried to determine the oligomeric state of purified LRRK2 with any of the quantitative biophysical methods that were used for the CtRoco protein. Interpretation of all these results was hindered by the high heterogeneity of the protein. A summary of the results of these attempts is given below, for the information of the referee.

1) Negative stain electron microscopy

As a first approach, LRRK2 was purified in the presence of GDP or GppNHp and directly after purification spotted on carbon-coated grids. As shown in Reviewer Fig. 5, the sample is very heterogeneous despite the seemingly high purity on SDS-PAGE (Reviewer Fig. 4). This heterogeneity could be explained by (i) conformational flexibility, (ii) different phosphorylation states of the protein, (iii) occurrence of degradation during and after purification, and (iv) aggregation. This

heterogeneity, showing a wide range of particle sizes, makes it very difficult to distinguish monomeric from dimeric protein.

Reviewer Fig. 5: Negative stain EM images of LRRK2 purified in presence of either GDP or GppNHp. LRRK2 was applied on a carbon-coated copper grid at a concentration of 0,05 mg/ml in the presence of 100 μ M GDP or GppNHp. After washing and staining with 1% uranyl formate, the grids were visualized under the microscope at a magnification of 60 000x.

2) Immunostaining

As a second approach, immunostaining of LRRK2 was performed repeating the protocol of Civiero *et al*, 2012. In short, LRRK2 was spotted on a carbon-coated grid, followed by a 1h incubation with rabbit anti-Flag antibody. After 3 washing steps, the LRRK2-anti-Flag complex was incubated for 30 min. with anti-rabbit-gold (10nm) antibody. The grid was again washed 3 times and finally stained with 1% uranyl formate. As a negative control, this procedure was repeated without LRRK2 (but with addition of primary and secondary antibodies). Representative images in Reviewer Fig. 6 show that the number of gold particles observed on the grids was comparable with and without LRRK2 (in both cases equal amounts of primary and secondary antibody were used). In retrospect this can be explained by a direct adsorption of antibodies on the grids, implicating that the gold particles do not necessarily represent the presence of a LRRK2 particle.

This very high background of non-specifically adsorbed antibodies, in combination with the very heterogeneous LRRK2 sample, in our opinion makes this experiment unsuitable for a quantitative analysis with regard to LRRK2 monomer/dimer ratio's.

Reviewer Fig. 6: immunostaining of LRRK2. LRRK2 was spotted on a carbon-coated copper grid at a concentration of 0,01 mg/ml (left upper panel) or 0,05 mg/ml (left lower panel). The protein was incubated for 1h with anti-FLAG rabbit antibody. After three washing steps, the grid was incubated with anti-rabbit-gold (10 nm) antibody for 30 min. The grid was then again washed and finally stained with 1% uranyl formate. Micrographs were collected at 60 000x magnification. Representative images are shown (left panels). As a negative control the experiment was repeated without LRRK2 (right panels).

3) Native mass spectrometry

As a third approach, several attempts were made on different protein batches to perform native mass spectrometry on LRRK2 purified in the presence of either GDP or GppNHp. Very weak overall signals were obtained. As shown in Reviewer Fig. 7 the spectra either in presence of GDP or GppNHp show a very high complexity indicative for heterogeneity of the sample. Spectra were recorded with 100V and 200 V sample cone voltage to dissociate possible interaction partners.

Although differences are present between the samples purified in GDP and GppNHp, the obtained masses can't be attributed to any of the expected molecular masses ($MM_{\text{monomer}} = 292 \text{ kDa}$) making annotation very hard. Some of the peaks (e.g. at $\pm 730 \text{ kDa}$) disappear at 200V, while peaks at lower molecular mass (± 27 and $\pm 36 \text{ kDa}$) appear, suggesting that these might be due to interacting proteins (for example 14-3-3 protein ($\pm 29 \text{ kDa}$)). Conclusions concerning a monomer/dimer equilibrium can hence not be made here based on native MS.

Reviewer Fig. 7: Native MS spectra LRRK2 in presence of GppNHp or GDP, collected at two sample cone voltages (100V and 200V). Although spectra for GDP and GppNHp LRRK2 could be recorded, the samples prove to be very heterogeneous. Only approximate experimental masses could be assigned to the peaks. Data quality is too low to draw any conclusions.

Minor comments:

The presentation of the EM-data require optimization. The negative stain images are badly recognizable and a discrimination of different dimerization states on their basis is hardly possible for the unfamiliar reader.

It is indeed challenging to show these images in a way that is easily recognizable for the unfamiliar reader. This is in part because we have been trying to reach a consensus between zooming out to show sufficient particles on one image to be trustworthy, and zooming in on the individual particles to make them easily interpretable.

In the revised manuscript we have improved this by:

- In Fig 2f (previously Fig 1g) we zoom in on a number of representative (boxed) particles on the EM images. We also included class averages of CtRoco in the GDP-bound state.
- In Fig. 3 (previously Fig. 2) we zoom in more on the individual images.
- We repeated the time-resolved single-turnover EM and show the clearest images in Fig. 4 (previously Fig. 3). Representative images of both experiments are shown in Supplementary Fig. 9.

Reviewer 3:

These results are undoubtedly interesting and would have been worth publishing.

We thank the referee for this positive comment.

The problem is however that, as presented, the experimental SAXS data do not appear to fully agree well with the conclusions of the authors. Appropriate revision of the Ms is therefore required.

*1) Indeed, Figure 1a,e depict the six relevant X-ray scattering curves displaced in logarithmic scale. Looking at Figure 1a, the CtRoc-COR GDP curve (green) is admittedly essentially parallel to the NF indicating that addition of GDP does not change the overall shape. However, the discrepancy of the fit for CtRoc-COR GDP (Chi=6.4) is very large not only compared to the NF (Chi=1.1) but also compared to CtRoc-COR GppNHp (Chi=3.3), which is not understandable, as the latter is supposed to be a monomer. Overall, do the authors report Chi or Chi**2 values?*

We thank the referee for pointing this out. We agree that comparison of the Chi values would give the impression that the GDP-bound form of CtRoc-COR is more different from the crystal structure of nucleotide-free CtRoc-COR compared to the GppNHp-bound CtRoc-COR. However, as the referee also points out a visual inspection already shows that the GDP-bound state and the nucleotide-free state are essentially parallel and match better with the theoretical scattering curve of the structure (grey) as compared to the GppNHp bound state.

This discrepancy was caused by the fact that the SAXS data for the different nucleotide-bound states of CtRoc-COR were collected using different X-ray sources (SWING at Soleil versus BM29 at ESRF). Because experimental errors are estimated differently at different beamlines, comparing the absolute Chi-values for data collected at different beamlines can be deceptive. To overcome this rather technical problem, we re-collected all CtRoc-COR SAXS curves under the same conditions at the SWING beamline. We now also report the more commonly reported Chi² value, rather than Chi values.

The Chi² values are now in agreement with our conclusions that CtRoc-COR in the nucleotide-free state is dimeric and resembles very well to the crystal structure (Chi² = 0.9), in the GDP-bound state CtRoc-COR is also mainly dimeric in SAXS (Chi² = 1.6), while in the GppNHp-bound state CtRoc-COR exists in a monomer-dimer equilibrium (Chi² = 6.0).

To emphasize the visual difference, Supplementary Fig. 2e is added where the three scattering curves are placed at the same intensity off-set.

2) For the CtRoco, the situation is yet more worrisome. Contrary to what the authors write (lines 113-115), in Fig. 1e, the green and red curves (corresponding to GDP and GppNHp bound states) are essentially parallel to each other and they both noticeably differ from the black curve (NF). From this plot it follows that the GDP and GppNHp states are similar to each other and differ from NF. Further, in Supplementary Table 2, the radius of gyration of CtRoco GppNHp (6.5 nm) is close to that of CtRoco GDP (6.2 nm) and both significantly exceed the Rg of CtRoco MF (5 nm). The authors must explain this controversy.

This is a very good point and we thank the referee for pointing this out. It is correct that although our MALS and SAXS data show that in the GDP state the protein is still mostly dimeric at the concentration used, the SAXS data shows an extension of the Rg values. This is also observed in the normalized pair-distance distribution function (Supplementary Fig. 4d, repeated below). The P(r) analysis shows that when bound to GppNHp the average intramolecular distances in CtRoco decrease (indicating monomerization), while there is a tailing toward longer distances showing that the molecule is more elongated in the monomeric state. In the GDP-bound state the average intramolecular distance of CtRoco is closer to the nucleotide-free state in agreement with a dimeric

protein, but also here tailing toward longer distances is observed indicating also some conformational changes in the dimeric GDP-bound state. Since this increase in R_g and the tailing in the $P(r)$ plot is not observed for GppNHp- and GDP-bound CtRoc-COR, we interpret the elongation of nucleotide-bound CtRoco as a change in conformation of the LRR domains vis-à-vis the Roc-COR domain upon nucleotide binding (see also our answer to comment 2 of referee 4).

To improve visual comparison of the different scattering curves we also added Supplementary Fig. 4e (repeated below) showing the three CtRoco scattering curves at the same intensity off-set. This figure shows that the SAXS curve of GDP-bound CtRoco isn't parallel to the one of GppNHp-bound CtRoco, but rather is more similar to nucleotide-free CtRoco and lies in between the curves for nucleotide-free and GppNHp-bound CtRoco, showing features of both.

Supplementary Figure 4: (d) Normalized pair-distance distribution functions ($P(r)$) of CtRoco in nucleotide-free state (black) or bound to GDP (green) or GppNHp (red). The smaller average intramolecular distance in presence of GppNHp is indicated with the black rectangle. Tailing of the curves with GppNHp and GDP is indicated with black arrows. (e) Scaled SAXS curves of CtRoco in nucleotide-free state (black) or bound to GDP (green) or GppNHp (red).

We included the following discussion in the results section to address this point (p 7-8). “It might seem remarkable at first sight that despite the difference in molecular mass the nucleotide-free, GppNHp- and GDP-bound CtRoco proteins elute at approximately the same volume in SEC (Fig. 2d). Correspondingly, Guinier analysis of the SAXS data shows that the radius of gyration (R_g) even increases upon going from the nucleotide-free state, over the GDP-bound state to the GppNHp-bound state (Supplementary Fig. 4). This could be explained by a conformational change occurring in the CtRoco subunits upon monomerization or upon nucleotide binding. Indeed, as is the case for the CtRoc-COR construct, the SAXS pair-distance distribution function of CtRoco shows a shift toward shorter average inter-atom distances in the presence of GppNHp compared to GDP or nucleotide-free states, indicative of monomerization (Supplementary Fig. 4d). However, the curve of the GppNHp-bound CtRoco protein also shows a tailing toward longer maximal distances. This strongly suggests the occurrence of a GppNHp-bound monomer that is more elongated than the corresponding subunits in the nucleotide-free dimer. Also in the GDP-bound state tailing of the pair-distance distribution function is observed indicating that already some conformational changes occur in the dimeric nucleotide-bound protein. These conformational changes also explain why the observed monomer/dimer equilibrium would probably be missed when assessed only with size-exclusion chromatography.”

Finally, we now also included native mass spectrometry and analytical ultracentrifugation results in the revised manuscript (See new Figures 1 & 2), which further corroborate the dimer-monomer transition upon binding of GppNHp.

3) The authors present *ab initio* shape for CtRoc-COR just to show that it overlaps with the crystal structure. Why do they not provide the shapes of CtRoco, which would have been much more interesting?

We didn't generate this envelope in first instance since there is no crystal structure of full length CtRoco to compare it with. However, as suggested by the referee, we calculated an *ab initio* envelope of the nucleotide free CtRoco protein and included it in the manuscript (Figure 2b, new manuscript numbering). Comparing the two envelopes (CtRoc-COR and CtRoco) with the dimeric CtRoc-COR crystal structure reveals extra density in the CtRoco envelope that can be attributed to the missing N-terminal LRR and C-terminal domain (see also Reviewer Fig. 8 for a superposition of both envelopes).

Reviewer Fig. 8: Superposition of the dimeric nucleotide-free CtRoc-COR crystal structure with the envelopes calculated from SAXS data of nucleotide-free CtRoc-COR (grey surface) and nucleotide-free CtRoco (black mesh).

4) In lines 119-120 the authors substantiate the presence of dimers by "relatively high protein concentration used in these experiments (8 mg/ml)". This is actually the initial solute concentration -- is the protein not substantially diluted on the column?

For the SEC-SAXS experiments we injected 50 μ l of an 8 mg/ml protein solution on the SEC column. Estimation of the protein concentration in the elution peak from the absorbance shows that the protein gets approximately 4 times diluted at the peak maximum.

In the SEC-MALS experiments even a smaller volume (10 μ l) of the protein was injected on the column.

5) The motivation and the gain for the authors to conduct in-line SEC/SAXS experimnt is not clear. These experiemnts, which are much longer and more complicated than standard SAXS measurements, are conducted if the samples are notoriously polydisperse and cannot be purified to have clean solutes for the batch analysis. In the SEC data displayed in the Ms, single peaks are shown, and it was not indicated what was the advantage of SEC/SAXS in this case.

We routinely use SEC-SAXS as standard setup since this was giving us also the best results in the past. The rational is that SAXS is extremely sensitive to small amounts of aggregates, even amounts that are barely noticeable on a chromatogram. The SEC step will make sure that these small amounts of aggregates are removed just prior to collecting data. Moreover, the coupling to SEC also automatically provides a perfect buffer match by collecting data in the void volume. Finally, by collecting data throughout the peak also data at different protein concentrations is obtained.

Moreover, for CtRoco GDP/GppNHp samples, the authors speak about dimer/monomer mixtures, (where SEC/SAXS should indeed be useful) -- why are these mixtures not separated at all on the column?

At the concentration used for the SEC-SAXS analysis, the GppNHp-bound CtRoco is indeed in a monomer-dimer equilibrium. We believe that the monomers and dimer are not separated because separation of macromolecules via size exclusion chromatography is based on their Stokes radius and not on molecular weight. Consequently, an elongated protein can elute at the same volume as a globular protein with twice its molecular weight (see for example Harold P. Erickson, 'Size and Shape of Protein Molecules at the Nanometer Level Determined by Sedimentation, Gel Filtration, and Electron Microscopy', *Biological Procedures Online*, 11.1 (2009), 32–51). We believe that this is what we observe for the CtRoco protein. We hypothesize that nucleotide-induced monomerization is accompanied by an elongation of the individual subunits, which results in roughly the same elution volume of the monomeric and dimeric proteins. Clear indications for such an elongation are provided by SAXS analysis that shows that the R_g and D_{max} values of nucleotide-bound CtRoco are larger than the values of nucleotide-free CtRoco. Also EM shows more elongated "worm like" particles for GppNHp-bound CtRoco as compared to the X-shaped particles for the nucleotide free protein (see also our answer to the referee's remark 2).

Minor comments:

1) It appears that there are some mistakes in Refs 42, 52, 54.

We thank the referee for pointing this out. The references (now references 44, 54 and 56) were corrected.

2) Numerous times in the text and supplementary, the authors write "scatter curves" -- they should use "scattering curves".

This was corrected.

3) Finally, SAXS data/models should be properly deposited (www.sasbdb.org) and the accession codes reported in the publication.

The SAXS data is submitted to the SASBDB and the accession ID's are reported in the paper. We will release the data prior to publication.

Reviewer 4:

In their manuscript, Deyaert et al. use SAXS, MALS, FRET and EM to describe Roco upon treatment with nucleotides and their analogue in comparison to the apo-form. Based on their data, the authors suggest a model for a monomer-dimer transition of the bacterial Roco protein dependent on the nucleotide state. The data presented are, however, not sufficient to differentiate between the suggested monomer-dimer transition and a conformational rearrangement. Furthermore, the EM images shown are of poor quality in particular upon nucleotide treatment.

Major points:

1. Upon addition of GDP, GppNHp, or GTP, Roco/CtRoco aggregates (e.g. Fig. 1g; 2c, 3c).

As further substantiated under remark 2, we do not agree with the referee that the protein aggregates upon addition of GDP, GppNHp, or GTP. We agree that it is difficult to represent the EM images in a way that is easily interpretable. This is in part because we have been trying to reach a consensus between zooming out to show sufficient particles on one image to be trustworthy and zooming in on the individual particles to make them easily interpretable.

In the revised manuscript we have improved this by:

- In Fig 2f (previously Fig 1g) we zoom in on a number of representative (boxed) particles on the EM images. We also included class averages of CtRoco in the GDP-bound state.
- In Fig. 3 (previously Fig. 2) we zoom in more on the individual images.
- We repeated the time-resolved single-turnover EM and show the clearest images in Fig. 4 (previously Fig. 3). Representative images of both experiments are shown in Supplementary Fig. 9.

The GppNHp state appears to adopt a less compact, elongated, “worm”-like shape. These samples do not appear to be just “half” of the nucleotide state. Along these lines, the authors also state that “the radii of gyration ... do not change much upon nucleotide binding suggesting that the protein subunits become elongated in their monomeric form”. Furthermore, the authors admit the “occurrence of secondary inter-domain conformational changes in the CtRoco monomers compared to the dimers”. The data presented are, however, not sufficient to differentiate between the suggested monomer-dimer transition and a conformational rearrangement and further experiments are needed to clarify these points. In order to provide further support for a dimer-monomer transition suggested by the authors, 3D structures calculated from the SAXS and EM data as well as immuno-electron microscopy (using antibodies that allow differentiation between the monomeric and the dimeric state) coupled to statistical analysis are required.

In addition to the SEC-MALS data that was included in our original manuscript and which shows that GTP/GppNHp binding shifts the molecular mass toward a lower value (note that MALS is independent of shape), we included three other techniques to support our conclusion.

- 1) Native Mass spectrometry performed on both CtRoc-COR (new Fig. 1e) and CtRoco (new Fig. 2e) shows that these proteins are nearly exclusively dimeric in the nucleotide-free state. Upon addition of GDP a very small amount of monomer is observable. Finally, when bound to GppNHp a nearly 50/50 monomer/dimer equilibrium is observed, in good agreement with SEC-MALS and SEC-SAXS data (MM obtained for CtRoc-COR from native MS: $MM_{\text{dimer}} = 130145$, $MM_{\text{monomer}} = 65069$; MM obtained for CtRoco from native MS: $MM_{\text{dimer}} = 256086$,

$MM_{\text{monomer}} = 127774$). An additional advantage of native-MS is that monomer and dimer species can be separated and observed.

- 2) Sedimentation velocity analytical ultracentrifugation (SV-AUC) experiments on CtRoc-COR (new Fig. 1c and Supplementary Fig. 3) and CtRoco (new Fig. 2c and Supplementary Fig. 5) further corroborate these results. For the nucleotide free and GDP-bound proteins a molecular mass very close to the dimer is observed, while in the GppNHp-state a molecular mass close to the theoretical MM of the monomer is observed.
- 3) The above experiment is even further corroborated by sedimentation equilibrium analytical ultracentrifugation (SE-AUC) experiments. These experiments have only been performed using full length CtRoco. While in the nucleotide-free state only a dimer is observed, an equilibrium is observed when bound to GDP and GppNHp. For the GDP-bound state an approximate monomer-dimer equilibrium constant of $2 \mu\text{M}$ is found. For the GppNHp-bound state an approximate monomer-dimer equilibrium constant of $30 \mu\text{M}$ is found. These results are in good agreement with the general trend of all our other data.

We are very enthusiastic about these new data and we feel that inclusion of these data in our manuscript has further strengthened our conclusions.

As requested by the referee we also calculated a SAXS-based *ab initio* shape reconstruction for the nucleotide-free CtRoco protein. This envelope is included in the manuscript as Figure 2b. Comparing the two envelopes (CtRoc-COR and CtRoco) with the dimeric CtRoc-COR crystal structure reveals extra density in the envelope of CtRoco that can be attributed to the missing N-terminal LRR and C-terminal domain. Since CtRoc-COR and CtRoco bound to GDP or GppNHp occur as a monomer/dimer mixture, a SAXS envelope would have no physical meaning.

Finally, the referee also suggests to make 3D reconstructions based on EM of all protein species. The method of choice for doing this would be cryo-EM. Although potentially interesting, calculation of such 3D reconstructions would represent an entirely different study. We would therefore like to argue that this falls outside the scope of the current paper, which presents the Roco monomer/dimer cycle as a main message.

2. Why did the particles aggregate upon treatment (e.g. Fig. 1g, middle and right, upper panel; Fig. 2c and 3c)? The strong aggregation makes it challenging to judge the shape of the particles, and accordingly, the authors also did not provide any class averages for GDP state and the time-courses they present. Class averages for all samples are required.

As suggested by the referee we added representatives of class averages of GDP-bound CtRoco as Fig. 2f (previously Fig. 1g). All class averages are also shown in Supplementary Fig. 7. Visual inspection of Supplementary Fig. 7 shows that the class averages of GDP-bound CtRoco mainly consist of particles that either also occur in the set of class averages of nucleotide-free CtRoco (dimers, indicated with a black square in Supplementary Fig. 7b) or in the set of class averages of GppNHp-bound CtRoco (monomers, indicated with a black square in Supplementary Fig. 7b). This shows that GDP-bound CtRoco mainly exists as a monomer/dimer mixture at the low protein concentrations used in EM. However, some class averages within the set of GDP-bound CtRoco cannot be attributed to either nucleotide-free or GppNHp-bound CtRoco in a straightforward way (class averages not indicated by a square in Supplementary Fig. 7b). These might represent conformations that are unique to the GDP-bound state of CtRoco, as also explained in the paper.

We did not include class averages corresponding to all the time points of Fig. 3 and Fig. 4 (new figure numbering). These figures represent a time dependent conversion of dimers to monomers. As such class averages will just yield representatives of both these species. Therefore, we would like to argue

that calculating class averages will not yield sufficient added value to warrant the enormous labour that would need to be invested to obtain such class averages (picking of about 10000 particles per time point).

As already discussed under the first remark by referee 4, we do not agree that the EM images of GDP- or GTP-bound CtRoco represent aggregated protein. To clarify this we have improved the quality of our EM images, and in Fig. 2f we also added zoomed-in images on individual particles. Moreover, such aggregation upon mixing with GDP, GTP or GppNHp would be easily spotted by any of the other biophysical techniques that we have been using (SAXS, MALS, native EM, AUC). This is not the case and rather we see a conversion to monomeric species. Finally, Fig. 4c shows that upon rapid mixing with GTP the X-shaped dimeric particles are converted into monomeric “worm-like” particles that convert again into dimeric X-shaped particles after hydrolysis of GTP. If the worm-like particles would correspond to aggregates as suggested by the referee it would be very hard to imagine how they would untangle again in a time-dependent way after GTP hydrolysis.

To convince the referee that the particles we observe in EM can account for non-aggregated monomeric and dimeric CtRoco we generated theoretical 2D EM projections of structural models using the program EMAN2 (see Reviewer Fig. 9). Since structures of full length Roco protein are not available, we first generated a hypothetical model of a *C. tepidum* LRR-Roc-COR protein by superimposing the available structures of the CtLRR domain (5IL7) and the CtRoc-COR domain (3DPU) using a common helix present in the structure of both the LRR domain and Roc-COR domain (Reviewer Fig. 8a) Please note that a very similar arrangement of the LRR-Roc-COR was validated by SAXS in Guaitoli *et al.*, PNAS 2016, and used in that paper to start building a model of LRRK2. When we generate theoretical 2D EM projections of this dimeric model it is clear that they are in good agreement with our experimental EM class averages of the nucleotide-free CtRoco protein (Reviewer Fig. 9a, right panel). However, for completeness it should also be noted that CtRoco used in EM contains an additional C-terminal domain of 150 amino acids that is not present in the structure model. Next we created two even more hypothetical models of a CtRoco monomer: one by just taking half of the dimer (Reviewer Fig. 9b); and the other by additionally rotating the LRR domain around the common helix toward the most elongated conformation, since an elongated monomer is observed in SAXS (Reviewer Fig. 9c). Comparison of the theoretical 2D EM projections of these models with representatives of the experimental “worm-like” EM class averages of GppNHp-bound CtRoco shows a good resemblance. Not all theoretical projections are observed in EM, which can be accounted for by a preferred orientation of the proteins on the grid, as commonly observed. This again shows that our EM images can indeed represent a monomeric form of CtRoco.

We would like to stress that we realize that the above treatment is a handwaving approach, which we merely use here to show to the referees that the EM particles we obtain can indeed account for a monomeric and dimeric form of CtRoco and as such do not correspond to aggregates. High resolution structural information of the proteins bound to different nucleotides would be required to show the details of these monomeric and dimeric particles. However, we would like to argue again that this is outside the scope of our paper, which focusses on the nucleotide-induced monomer/dimer equilibrium.

Reviewer Fig. 9: Comparison of theoretical 2D EM projections of a model of the LRR-Roc-COR domains of CtRoco with experimental negative stain EM class averages of CtRoco. (a) Model of dimeric CtLRR-Roc-COR based on alignment of a common helix of the CtLRR (yellow) and CtRoc-COR (Roc domains in red, COR domains in blue) structures. (b) Monomer of the model shown in (a). (c) Elongated arrangement of a monomer by rotating the LRR domain around the common helix. The middle panels show theoretical 2D EM projections of the corresponding models. The right panels show an alignment of 5 theoretical 2D EM projections with 5 experimental class averages of nucleotide free CtRoco (a) or GppNHp-bound CtRoco (b) and (c).

3. Fig. 1g, GppNHp: How do the authors explain the “extra” densities in the class averages? How do the authors exclude that these are rearranged particles?

We would like to refer here to our answer to the previous point (Reviewer Fig 8). The theoretical 2D EM projections of monomeric CtRoco particles also show the “extra densities” that the referee refers to. This means that they can be accounted for by the way the LRR and Roc-COR domains are oriented toward each other.

4. Fig. 3c “CtRoco is almost completely converted from a dimeric to a monomeric form after 240 seconds.” “The EM images do not support this claim. Instead, severe aggregation is observed.”

As already argued above we do not agree that the sample is aggregated. We also repeated this time resolved experiment with a second batch of protein. Both repeats of the experiments are now included in the document (Fig. 4 and Supplementary Fig. 9).

5. L. 82: *Ab initio* models based on the SAXS data should be provided for all data sets as listed in Suppl. Table 1-3.

We included an *ab initio* model based on SAXS of nucleotide-free CtRoco in Fig. 2b. Since CtRoco and CtRoc-COR in their nucleotide bound states exist as a monomer/dimer equilibrium the SAXS envelopes would have no physical meaning. Therefore we did not include these envelopes.

6. Models need to be deposited in a relevant database such as <https://www.sasbdb.org/> and accession numbers indicated in the manuscript

The SAXS data is submitted to the SASBDB and the accession ID's are reported in the paper. We will release the data prior to publication.

7. L. 82: Delete “very good agreement” and provide additional data. The fit looks rather random. How was the crystal structure fitted into the *ab initio* model? Are there alternative fits and which support do the authors have to exclude the alternative fits?

We deleted “very”. For placing the structure in the SAXS envelopes we have used the program Supcomb from the ATSAS suite, which is routinely used for this purpose. In light of a remark made by referee 2 we also remeasured the SAXS data for CtRoc-COR on the SWING beamline rather than the BM29 beamline. Although the results and the corresponding envelopes are essentially the same, the visual fit to the envelope seems even better based on these data. To show the goodness of this fit we now show the envelope in two orientations (Fig. 2b).

Finally, we would like to note that the most objective way to compare an X-ray crystallography model with its SAXS curve is the direct correlation of the experimental scattering curve and the theoretical scattering curve of the crystallographic model. Figure 1a shows a χ^2 value of 0.9 for the comparison of the nucleotide-free CtRoc-COR SAXS curve and the dimeric CtRoc-COR crystal structure. Because χ^2 values between 0.9 and 1.1 are generally accepted as very good, this validates the crystallographic model.

8. L. 343ff: The description of the EM methods does not provide sufficient information to allow the reader to judge whether or not the data are sound. How did the authors exclude that the treatment (e.g. washes and drying steps) influence the shape of the particles?

We used a standard staining protocol that lowers the chance of staining artefacts as described in Booth et al. *Journal of Visualized Experiments*, 2011, 1–8 and Rames et al. *Journal of Visualized Experiments : JoVE*, 2014, e51087.

Moreover, Reviewer Fig. 8 shows that the shapes of the particles we observe generally correspond to what is expected for dimeric or monomeric CtRoco.

9. L. 417, ref. 21: The authors should discuss this reference in detail. A dimeric state of LRRK2 was found in the presence of GppNHp in ref. 21 – how do the authors explain these findings? How does the data presented in ref. 21 fit to the authors' model? The authors should also prepare a figure, where they compare their model and the previously published model (ref. 21).

The Roc-COR supradomain is highly conserved among all Roco proteins, including LRRK2. As such it can be expected that the main function of this domain is also widely conserved. Similar to the prokaryotic Roco proteins, various studies report that LRRK2 can form dimers through its COR domains *in vitro* (e.g. Terheyden *et al.*, *Biochem. J.* 2014 and references given therein). However, other studies suggest that LRRK2 is a monomer or exists as a monomer/dimer equilibrium *in vivo* (Nixon-Abell *et al.*, *Front. Mol. Neurosci.* 2016; Ito & Iwatsubo, *Biochem J.* 2012; James *et al.*, *Biophys J* 2012, ...). These findings agree with our results and can now be re-evaluated in light of the results presented in our paper. Of course, it should be noted that apart from the LRR, Roc and COR domains, LRRK2 contains several other domains which could contribute to the dimer interface and which could make the mechanism even more complicated in LRRK2.

In Ref.21, the influence of adding nucleotides on LRRK2 oligomerization was assessed using blue native PAGE. No shift of the bands was observed upon addition of nucleotides. In our opinion these

experiments are not conclusive. First, submitting the protein to an electrical field in gel electrophoresis might have a big impact on the nucleotide load, since the negatively charged nucleotides might be pulled out of the protein's active site. Moreover the nature (monomer/dimer) of the band observed on native PAGE is still under debate (see Ito, G. & Iwatsubo, 2012). Indeed, we also observe that the monomeric and dimeric form of CtRoco cannot be distinguished using a separation method that heavily relies on the shape of the particle, such as size-exclusion chromatography or blue native PAGE. In our case this is due to the elongation of the monomer as discussed on page 7 of the manuscript.

Due to the technical difficulties in working with LRRK2 (low expression, instability, aggregation) it has proven so far very difficult to prove or disprove that the same nucleotide-dependent monomer/dimer equilibrium exists for LRRK2 using any of the quantitative biophysical methods that we use in our paper. We would like to refer to our answer to referee 2 for details.

We discuss these points now in more detail in the conclusions of our manuscript, and the following paragraph was included (p 14-15): "Since the presence of the Roc-COR module is highly conserved throughout all Roco proteins, our current findings also throw new light on LRRK2 functioning. However, apart from the Roc-COR and LRR domain, LRRK2 also contains a kinase domain and a number of domains typically involved in protein-protein interactions, which potentially can have an additional influence on oligomerization^{33,34}. Detailed and quantitative biophysical studies with LRRK2 to determine its oligomeric state remain highly challenging since purification yields are low, and purified LRRK2 is very heterogeneous and prone to degradation. However, it has been shown that LRRK2 can form dimers via its COR domain^{15,20,21,23,35}, while other studies indicate that LRRK2 can also exist as a monomer^{18,20,24}. Both observations can be reconciled with our model. Interestingly, recent data suggest that LRRK2 can exist as either a monomer or a dimer in the cell, where the monomeric form prevails in the cytosol and the dimeric form is more prominently present at the membrane^{25,26,36,33}. A higher prevalence of a monomeric GTP-bound form under multiple GTP turnover conditions such as present in the cytosol (GTP concentrations in the cell are in a range of 1-1.6 mM^{37,38}) corresponds to our *in vitro* observation in EM (Fig. 3). Following the same argument, LRRK2 would exist in a more GDP-bound form at the membrane. Apart from these considerations it is likely that additional regulatory factors, such as phosphorylation or interaction with other proteins, influence the monomer-dimer equilibrium of LRRK2 *in vivo*³⁹."

Minor points:

10. L. 31-32: Delete "thereby potentially opening new avenues towards drug design". The manuscript does not provide data on drug design.

This was deleted.

11. L. 234: Delete "for the first time unequivocally".

This was deleted.

12. L. 290: Please provide more details for "lung and kidney abnormalities".

We changed "abnormalities" to "morphological changes". We cite references Herzig *et al.*, 2011; Baptista *et al.*, 2013 and Fuji *et al.*, 2015 in our manuscript, where these effects are described in more detail.

13. L. 291ff: Delete this sentence as the manuscript does not provide sufficient support.

We deleted this sentence.

14. L.354: How many particles were averaged per class in average?

For nucleotide-free CtRoco 11571 particles were used to generate class averages (296 classes, so on average 39 particles per class). For GDP-bound CtRoco 9620 particles were used to generate class averages (130 classes, so on average 74 particles per class). For GppNHp-bound CtRoco 11164 particles were used to generate class averages (239 classes, so on average 48 particles per class).

15. L. 399: A space is missing in "thatstimulates".

This was corrected.

Reviewers' comments:

Reviewer #1 (Remarks to the Author):

The revised manuscript is much improved and clarified some of the reservations I expressed in the initial review. The body of work is publishable; however, I am not sure whether or not the significance of the findings presented in the manuscript fit Nature Communication, I will refer to the Editor for making that judgement. While understanding the structure and function of CtRoco is interesting, there is no evidence presented, data or literature referenced, to suggest that the findings presented would provide insights into the functioning of Parkinson's disease-associated protein LRRK2, which was the focus of the paper.

Additional comments that I hope that the authors would find helpful in revising their manuscript:

1. Page 3, line 55. "...the now widely adopted model that LRRK2 functions as a GAD..." I am not aware of it being widely adopted and I personally don't see it that way.
2. Page 3, lines 58-59, and page 15. "...LRRK2 can also form dimers through its COR domain in vitro Ref# 15,20-23." I am not aware of any evidence showing that LRRK2 forms dimers via COR-COR interaction. I have also read the 4 papers referenced, but none of them provided any data supporting that.
3. Page 4, line 73. "...setting a new paradigm for their GTP hydrolysis mechanism..." I don't see that in this manuscript.
4. Page 6. SEC-MALS data showing about 20% error range in estimating the molecular weights seem quite high for the technique, which is typically about 5-10%.
5. Page 5, lines 113-122. It is a little confusing to see on the same page "...completely shifted to the monomer..." and "...approximately 50:50 monomer/dimer ratio..." both under the same condition of GppNHp. More clear explanation might help.
6. Page 7, line 125. "...GppNHp induces monomerization in a concentration dependent manner." I didn't find any data supporting this statement.
7. Page 8, lines 160-162. "...in the GDP-bound state...indicating that already some conformational changes occur in the dimeric nucleotide-bound protein." This is confusing because Fig. 2D shows that the GDP-bound state is more compact than the Apo nucleotide-free conformation.
8. Page 10, line 219. "catalytically relevant time-scale" needs a definition.
9. Page 11, line 226. Please check to make sure that you indeed used "reversed-phase HPLC". We had to use a HILIC column on HPLC, which is normal-phase.
10. Page 12-13. I am confused on how the LRRK2 PD-associated mutations are mapped onto the structure of ct-Roco because there is no sequence homology at these sites.
11. Page 15, line 311-315. " We show here...that GTP binding induces monomerization..." this has been shown for LRRK2 before (Liao et al. PNAS 2014).
12. Fig. 1d, MW calculation range for the red curve is large and decreasing, suggesting that if the range is kept under more narrow, the MW determined might be smaller than the 78 kDa reported.
13. Fig. 1e and Fig. 2e: mass-spec curves should show intensity on the y-axis instead of % (% of what?).

14. Fig. 2f: The volume of density of the worm-shaped structures (supposedly monomers) are similar to that of the x-shaped structures (supposedly dimers). One does not look like two times the size of the other as one would expect if they are dimers and monomers of the same protein.

15. Fig. 4c: There is a lot of x-shaped structures throughout the time range, thus not a clear case of dimer converting to monomer.

16. Fig. 6: It is a little confusing as to under what circumstance would CtRoco exist in an apo nucleotide-free form as shown at the top of the figure.

Reviewer #2 (Remarks to the Author):

The authors have done an excellent job in addressing my comments. My main point of criticism was a potential lack of significance of the data in the context of related eukaryotic proteins (e.g. LRRK2). However, due to the response of the authors and in the light of the other referee comments I would now favor publication of this study in Nature Communications.

The authors therefore have replied to my comments to my full satisfaction.

Reviewer #3 (Remarks to the Author):

In my view the authors adequately responded to the criticisms and the revised Ms can be recommended for publication.

Just a small editorial comment: discrepancy is everywhere depicted as "Chi" -- the appropriate greek letter should be used instead.

Reviewer #4 (Remarks to the Author):

In their revised manuscript, Deyaert et al. added data to support their claim of a monomer-dimer transition of the bacterial Roco protein dependent on the nucleotide state. The major concern this reviewer has is related to the EM part of the study, which in the reviewer's view does not exclude a conformational change. In particular, there seems to be a discrepancy between the theoretical model (used to calculate theoretical EM projections) and the experimentally observed EM class averages. The experimental class averages appear smaller despite the fact that the theoretical model lacks 150 C-terminal amino acids (see Reviewer Fig. 1c). The concern is further substantiated by the Reviewer Fig. 9 a and b, where e.g. the third class average ("Experimental class averages (nucleotide free)") in the dimeric model (Reviewer Fig. 9a) fits better to the overall appearance of the theoretical 2D projections of the half model (second and third projection view, third row; Reviewer Fig. 9 b).

The model provided in Reviewer Fig. 9 c appears to be arbitrary, and does not exclude a conformational change of the protein. Did the authors test further conformational states?

The immunolabeling experiment in Reviewer Fig. 6 is not designed in a way that allows conclusions. Using anti-rabbit IgG(?) -10 nm gold is not appropriate for such a small particle. Furthermore, the very high background of non-specifically adsorbed antibody can be improved by using appropriate protocols.

Furthermore, it remains unclear whether or not the EM preparation (washing and drying followed by negative staining) could interfere with the conformation and/or nucleotide state.

Reviewer #5 (Remarks to the Author):

The manuscript 'A bacterial homologue of the Parkinson's disease-associated protein LRRK2 undergoes a monomer-dimer transition during GTP turnover' from Deyaert et al. reports a novel insight in the activation cycle of the kinase Roc-COR. To understand the function of the monomer-dimer transition and the *in vivo* significance of the protein is definitively important. The paper is well written. Nevertheless, I'm not convinced by the interpretation of the given results.

The author try to use a bunch of different methods, SEC-MALS, SEC-SAXS, FRET and EM to understand the conformation and function of CtRoco dependent on his nucleotide state. After changing the manuscript due to given Reviews comments, the results coming from MALS, SAXS and FRET are clearly to understand. The authors are showing a shift of the molecular mass of the protein due to different nucleotide states. The explanation why the dimeric protein with 252kDa elutes at the same volumes as the 173kDa protein (Fig2d) could be explained by a structural change of the monomeric protein. These conformational changes can be postulated on the basis of FRET experiments (Fig 3b).

Nevertheless, the basis of all this evaluations should be a well understood sample. The EM images are due to the addition of nucleotides GTP, GDP and GppNHp of poor quality and I'm not convinced that the images represent the turnover of a dimeric conformation to a functional monomeric protein. The images show clearly a degradation of the dimeric protein after adding nucleotides, shown by an inhomogeneity of the protein shape and sizes in given images. The author concentrates only on structures which reflect possible monomeric and dimeric conformations. All the broken proteins in between are not discussed. There is no proof that the conformation change of the protein due to the addition of nucleotide is specific and do not reflect degradation products of the complex. The turnover experiment (Suppl. Fig 9) indicating a turnover of GTP cycle can as well indicate a destabilization and reorganisation of the complex by the given condition. The author should show that the protein is stable under the given conditions.

The class averages of the particles given in Fig.2 are not well defined and centred. They reflect sometimes a combination of more than one protein. I do not agree with the author validations, that the reported EM images in the manuscript reflect a conformational change, proofed by a comparison of the class averages with theoretical 2D EM projections given in Reviewer Fig. 1c. The theoretical EM images do not correspond very well to the experimentally observed class averages of the free dimeric CtRoco, even under the argument of missing 150C-terminal amino acids. The superposition of the crystallographic model to the given *ab initio* SAXS model is still not very informative.

To proof the nature of the protein conformation IEM should be performed. The argument, that the Immunogold labelling causes an unspecific background is not valid. There are enough possibilities to deal with this background and get rid of it (Blocking reagents like BSA, wash steps, usage of Protein A gold, higher dilutions).

I fully agree with the author, that the question if CtRoco undergoes dimer-monomer cycle is important, but I do not agree that the data clearly indicates that the monomeric state of LRRK2 exists and leads to a new model of monomer-dimer transition. The manuscript therefore is acceptable but needs additional work.

I would consider that add additional proof of the stability of the sample before and after adding nucleotide. Immunogold labelling should be performed. The interpretation of EM data should be done with purified and stabilized samples. *In vivo* data showing the abundance of monomeric-dimeric conformations, as well as the location of these proteins should be added.

Manuscript No.: NCOMMS-16-22070A

A bacterial homologue of the Parkinson's disease-associated protein LRRK2 undergoes a monomer-dimer transition during GTP turnover

Point-by-point response to referee remarks

Reviewer #1 (Remarks to the Author):

1. Page 3, line 55. "...the now widely adopted model that LRRK2 functions as a GAD..." I am not aware of it being widely adopted and I personally don't see it that way.

We agree with the referee that still different models exist to explain the GTPase mechanism of the RocCOR domain of LRRK2. This is for example reviewed in Nixon-Abell et al. (2016) *Biochemical Society Transactions* 44, 1625-1634, where either a "Ras small GTPase model" or a "GTPase activated by dimerisation model (GAD)" is proposed. Yet, we think that it is more than fair to say that the GAD model is well-known and quite wide-spread in scientific literature regarding the GTPase activity of LRRK2.

However, to be more cautious we deleted the words "now widely adopted" in the revised manuscript.

Finally, we would like to emphasize that we are not defending the GAD model in our manuscript. Rather our findings show that the GTPase mechanism is much more complicated than previously proposed in the GAD model and involves a monomer-dimer transition.

2. Page 3, lines 58-59, and page 15. "...LRRK2 can also form dimers through its COR domain in vitro Ref# 15,20-23." I am not aware of any evidence showing that LRRK2 forms dimers via COR-COR interaction. I have also read the 4 papers referenced, but none of them provided any data supporting that.

In the paper by Terheyden et al (2015) (Ref #15) co-immunoprecipitation experiments were performed in HEK-293T-cells where GFP-fused truncated LRRK2 constructs were co-expressed with FLAG-tagged LRRK2 constructs. It was found that FLAG-tagged LRRK2 RocCOR interacts with both GFP-tagged LRRK2 RocCOR and GFP-tagged LRRK2 COR. In contrast GFP-Roc is not co-purified with FLAG-tagged RocCOR, supporting the notion that COR is necessary for dimerization (see Figure 5 in Ref # 15).

We would also like to point out that we are not claiming that dimerization of LRRK2 would only occur via the COR domains. Indeed, other domains could also participate. Our finding that nucleotide binding to the CtRoco Roc domain influences dimerization in fact indicates that also the Roc domain is implicated in dimerization, either directly or indirectly via conformational changes that are transmitted to the COR.

Despite the results of Reference # 15 we adapted the text as follows: *"...LRRK2 can also form dimers through its Roc-COR domain in vitro "*.

3. Page 4, line 73. "...setting a new paradigm for their GTP hydrolysis mechanism...", I don't see that in this manuscript.

We agree with the referee that we do not suggest a detailed molecular mechanism for GTP hydrolysis by the Roco proteins in the strict sense of the word. However, we do firmly believe that the concept that these proteins cycle between a monomer and dimer state during hydrolysis is new and throws an entirely new light on their mode of action.

In order to decrease the strength of our claim, we have changed the exact wording into "and propose a new model for their GTP hydrolysis mechanism".

4. Page 6. SEC-MALS data showing about 20% error range in estimating the molecular weights seem quite high for the technique, which is typically about 5-10%.

When we calculate the errors on the MALS data for the dimeric nucleotide-free and GDP bound proteins we find values well within (or better) than the 5-10% range (see below). Such a calculation is not valid for the GppNHp bound sample since in that case we have a mixture between monomeric and dimeric species as explained in the text.

For the CtRocCOR (Fig 1c):

Theoretical MM dimer = 129.8 kDa

MM (NF) MALS = 120 kDa => error = 7.5 %

MM (GDP) MALS = 125 kDa => error = 3.7 %

As explained in the text the sample in the GppNHp state is in equilibrium between a monomeric and dimeric form. Therefore, a MM in between these two values, but closer to the monomer, is observed. Under these conditions direct comparison of the observed MM to that of a monomer or dimer is meaningless.

For the CtRoco (Fig 2c)

Theoretical MM dimer = 254.2 kDa

MM (NF) MALS = 246 kDa => error = 3.2 %

MM (GDP) MALS = 252 kDa => error = 0.9 %

Again, the sample in the GppNHp state is in equilibrium between a monomeric and dimeric form.

5. Page 5, lines 113-122. It is a little confusing to see on the same page "...completely shifted to the monomer..." and "...approximately 50:50 monomer/dimer ratio..." both under the same condition of GppNHp. More clear explanation might help.

This remark of the referee also relates to his next point concerning the concentration dependency of the monomer / dimer equilibrium.

For all the experiments performed on p6 the protein concentrations used are such that the CtRoc-COR is a dimer in the nucleotide-free and GDP-bound forms. However, for the GppNHp bound state the equilibrium is shifted toward the monomeric state. The extent of monomerization depends on the concentration of the protein that was used in the experiment. For the SV-AUC experiment a protein concentration of 4.6 μM was used. At this concentration the monomer-dimer equilibrium is nearly completely shifted to the monomeric state when bound to GppNHp. For the native MS experiments, the protein concentration prior to injection into the mass spectrometer was higher and between 12.5 and 16 μM . In that case a 50:50 ratio monomer/dimer is found. This indicates that the monomer/dimer equilibrium is dependent on the protein concentration, which in fact is to be expected according to the laws of thermodynamics. We would like to avoid a very quantitative interpretation of the concentration dependency since the actual concentration in the mass spectrometer is not known (only prior to injection).

6. Page 7, line 125. "...GppNHp induces monomerization in a concentration dependent manner." I didn't find any data supporting this statement.

See our answer to the previous remark.

7. Page 8, lines 160-162. "...in the GDP-bound state...indicating that already some conformational changes occur in the dimeric nucleotide-bound protein." This is confusing because Fig. 2D shows that the GDP-bound state is more compact than the Apo nucleotide-free conformation.

In our experience the peaks on the SEC chromatograms have a tendency to broaden with the age of the column and the number of runs per day. Therefore we would not put too much emphasis on the relation between peak width and conformation. In contrast, the values obtained from SEC-MALS and SEC-SAXS are independent of peak width and are a robust measure of molecular mass and conformation, respectively.

8. Page 10, line 219. "catalytically relevant time-scale" needs a definition.

We replaced "catalytically relevant time-scale" by "and occurs within the time frame of a catalytic cycle of GTP turnover".

A few lines above (line 213) we defined that the GTP turnover time is about 10 minutes.

9. Page 11, line 226. Please check to make sure that you indeed used "reversed-phase HPLC". We had to use a HILIC column on HPLC, which is normal-phase.

We did use a reverse phase (C18) column. Since we add tetra butyl-ammonium bromide to our samples and running buffer, the nucleotides become hydrophobic. The more phosphates the more tetra butyl-ammonium bromide is bound and the more hydrophobic they become.

10. Page 12-13. I am confused on how the LRRK2 PD-associated mutations are mapped onto the structure of ct-Roco because there is no sequence homology at these sites.

There is considerable sequence homology between all the Roc-COR domains within the Roco protein family (Bosgraaf et al., 2003). Supplementary figure 2 from Gotthardt et al., 2008 displays a multiple sequence alignment of the Roc-COR domains from different Roco proteins including the CtRoco protein and the human LRRK2 protein. This alignment shows some highly conserved motifs. Mapping the conserved residues onto the CtRoc-COR crystal structure shows that the interface between the Roc and COR domains is particularly conserved; all the PD analogous mutations localize exactly to this interface (Gotthardt et al., 2008).

The paper of Guitoli et al., 2016 presents a model of LRRK2 based on experimental restraints and homology modelling. Structural alignment of the CtRoc-COR crystal structure with the LRRK2 Roc-COR homology model from that paper, results in the same analogous PD mutations: LRRK2 I1317V aligns with CtRoco L487A, LRRK2 R1441C/G/H with CtRoco Y558 and LRRK2 Y1699C with CtRoco Y804C (Reviewer figure 1).

Reviewer figure 1: Sequence alignment of the CtRoco Roc-COR domain (AA454-876) and Hs LRRK2 Roc-COR domain (AA 1336-1783) resulting from a structural alignment of the Ct Roc-COR crystal structure (3DPU) and the Hs LRRK2 Roc-COR homology model (Guitoli et al., 2016). Overall a sequence similarity of 46.3% is found. Pathogenic PD mutation of LRRK2 and the analogues residues in CtRoco are indicated. Amino acid numbers of the CtRoco protein are shown.

11. Page 15, line 311-315. " We show here...that GTP binding induces monomerization..." this has been shown for LRRK2 before (Liao et al. PNAS 2014).

In this paragraph of the discussion (line 311-315) we explain that the structure of the *Ct* Roc-COR domain shows a clear dimer, mainly through interactions of the COR domains. Furthermore we say that it was previously proposed that this dimer would be strengthened in the GTP state by tightening the interaction between the Roc domains. The claim we subsequently make is "We show here, using exactly the same protein, that GTP binding induces monomerization and that GTP hydrolysis occurs after monomerization."

To the best of our knowledge it is correct to say that it is the first time that this is shown for a Roc-COR domain as well as for a full length Roco protein (in the reference mentioned by the referee an isolated Roc domain, lacking the COR domain, is used).

12. Fig. 1d, MW calculation range for the red curve is large and decreasing, suggesting that if the range is kept under more narrow, the MW determined might be smaller the 78 kDa reported.

We agree with the referee. The size exclusion chromatography peak in the presence of GppNHp corresponds to a monomer-dimer mixture, reflected in the gradually decreasing MM throughout the peak. This results in an average MM of 78kDa, in between the expected value for a monomer and dimer (although closer to the monomer). Calculating the MM at the “end” of the elution peak would result in a MM of 72 kDa, even closer to the expected MM of a monomer. However, we chose to use the MM at the peak average throughout for consistency.

13. Fig. 1e and Fig. 2e: mass-spec curves should show intensity on the y-axis instead of % (% of what?).

The % is a relative intensity that refers to the peak intensity in relation to the base peak (most intense peak) of the spectrum. To clarify this, we add “rel. intensity %” to the axis. This is a very common way to represent MS data.

14. Fig. 2f: The volume of density of the worm-shaped structures (supposedly monomers) are similar to that of the x-shaped structures (supposedly dimers). One does not look like two times the size of the other as one would expect if they are dimers and monomers of the same protein.

The 2D projections of the “worm-shaped” particles do appear somewhat larger than just half of the X-shaped dimers. We believe that this is due to the conformational changes within the protomers such that the particles become less compact. Similar conformational changes leading to less compact particles upon GppNHp binding are also observed by SAXS. Moreover, we have ample other evidence that CtRoco monomerizes upon GppNHp/GTP binding: SAXS, MALS, AUC, native MS and FRET.

15. Fig. 4c: There is a lot of x-shaped structures throughout the time range, thus not a clear case of dimer converting to monomer.

A significant amount of X-shaped structures are present at each time point during the single turnover experiment. This is to be expected since we follow the particle shapes upon mixing of 1 μ M GTP and 1 μ M CtRoco. These concentrations are only 2-fold above the reported K_D for GppNHp (0.5 μ M; Gotthardt et al, 2008) resulting in a certain amount of unbound protein. Moreover, since GTP is being hydrolysed during the reaction, a mixture of unbound, GTP-bound and post-hydrolysis GDP-bound CtRoco molecules will co-exist. However, when a large excess of GppNHp or GTP is used (1 mM), like in Fig. 3c, a full conversion to monomers is observed.

To clarify this in the manuscript, the text has been adapted and now reads as follows: "Under these single turnover conditions a large fraction of CtRoco molecules are converted from a dimeric into a monomeric form after 240 seconds. This is expected for a single turnover experiment where a mixture of unbound, GTP-bound and post-hydrolysis GDP-bound CtRoco molecules co-exist."

16. Fig. 6: It is a little confusing as to under what circumstance would CtRoco exist in an apo nucleotide-free form as shown at the top of the figure.

Considering the high cellular concentrations of GDP and especially GTP, the nucleotide-free form of the protein is not expected to be highly populated in a cellular context. However, it remains a necessary (probably very transient) intermediate in the catalytic cycle, since GDP needs to be released before GTP can bind. Purely based on kinetic considerations, the amount of monomeric versus dimeric protein would hence depend on the relative magnitudes of the rates of GTP binding, monomerization, GTP hydrolysis and dimerization (which in the GDP state depends on protein concentrations). As pointed out in the discussion, within the cell also many other factors could play a role: phosphorylation, membrane interactions, interactions with other proteins, etc.

Reviewer #2 (Remarks to the Author):

The authors have done an excellent job in addressing my comments. My main point of criticism was a potential lack of significance of the data in the context of related eukaryotic proteins (e.g. LRRK2). However, due to the response of the authors and in the light of the other referee comments I would now favor publication of this study in Nature Communications.

The authors therefore have replied to my comments to my full satisfaction.

We thank the referee for this positive response.

Reviewer #3 (Remarks to the Author):

In my view the authors adequately responded to the criticisms and the revised Ms can be recommended for publication.

Just a small editorial comment: discrepancy is everywhere depicted as "Chi" -- the appropriate greek letter should be used instead.

We thank the referee for this positive response. We adjusted Chi to the appropriate Greek letter in the manuscript, as requested.

Reviewer #4 (Remarks to the Author):

In their revised manuscript, Deyaert et al. added data to support their claim of a monomer-dimer transition of the bacterial Roco protein dependent on the nucleotide state. The major concern this reviewer has is related to the EM part of the study, which in the reviewer's view does not exclude a conformational change. In particular, there seems to be a discrepancy between the theoretical model (used to calculate theoretical EM projections) and the experimentally observed EM class averages. The experimental class averages appear smaller despite the fact that the theoretical model lacks 150 C-terminal amino acids (see Reviewer Fig. 1c).

The main message of our paper is that CtRoco undergoes a dimer/monomer cycle upon GTP binding and hydrolysis. In our manuscript we provide multiple pieces of evidence that this is indeed the case: SAXS, MALS, AUC, FRET and native MS. The native MS that was added in the previous revision even separates and directly shows the monomer and dimer population. In combination with all these highly redundant pieces of experimental evidence, we believe that it is more than fair to interpret the conversion of X-shaped particles to worm-shaped particles as a dimer/monomer transition (please also see further for a discussion on the immunolabeling experiments).

The comparisons between the theoretical projections and the experimental class averages are only qualitative and were merely made to demonstrate to the reviewers that the particles we are observing are compatible with the shapes of monomers and dimers, and are also not included in the paper or supplementary data. We are in no way meaning to overstretch these comparisons and make any conclusions regarding the fine details of the domain arrangement within the monomers and dimers. We agree with the reviewer that the theoretical projections appear bigger than the experimental class averages. We would like to stress that when making these projections, the contrast and resolution (in this case 30 Å) of the projections can be chosen. In this way, the theoretical projections can show more details of the structure than the corresponding experimentally observed particles. Moreover, in the figure the referee is referring to, we also compare theoretical projections calculated from a single particle with class averages representing the average of many particles. In these class averages, certain details of the single particles with a lower density get lost due to averaging. It is well known that specifically for small particles, which is the case here, alignment errors can be relatively high (Henderson R et al (2011) J Mol Biol. 413(5):1028-46) resulting in smearing of the borders of the class averages, which as result appear smaller. Indeed, when comparing the class average with the extracted single particles constituting a certain class, it can be seen that the class average only shows the “common” shape of all these particles combined. Some of the single particles are larger than the class average and correspond better to the size of the theoretical particle projection, as shown in Reviewer figure 2.

Reviewer figure 2: Comparison of a class average, the extracted particles from this class and a theoretical projection. (a) The third class average of Reviewer figure 1c in our previous answers to the referees. (b) The corresponding theoretical projection. (c) The corresponding particles extracted from this class average showing that the size of the single particles is in good agreement with the size of the theoretical projection.

The concern is further substantiated by the Reviewer Fig. 9 a and b, where e.g. the third class average (“Experimental class averages (nucleotide free)”) in the dimeric model (Reviewer Fig. 9a) fits better to the overall appearance of the theoretical 2D projections of the half model (second and third projection view, third row; Reviewer Fig. 9b).

As pointed out in our previous responses to the referees, we would like to emphasize that the comparison of these theoretical and experimental projections do not constitute a basis for any structural conclusions and was used in our earlier responses just to indicate that the observed projections do not contradict our conclusions. The theoretical projections were generated merely to show to the referee that the particles we observe in presence of GppNHp are not aggregated but can account for monomeric proteins. Focussing on individual class averages and projections is in our opinion stretching the interpretation too far. Indeed, a number of projections generated from the dimers and monomers look rather similar. However, we think it is more than fair to say that overall we find a better global resemblance of the particles in nucleotide-free state with the theoretical dimers and a better resemblance of the particles in presence of GppNHp with the monomers. We can also not exclude that at the very high dilution used in EM some monomers would be present even in the absence of nucleotides.

Finally, we would also like to stress once again that the EM is not our prime method to show that CtRoco monomerizes in presence of nucleotides. Indeed, this is already very clearly shown by native MS, MALS, AUC and SAXS. Our EM just offers a direct visualization of the particles in both states.

As suggested by this referee and referee 5, we also performed nanogold-labelling, see below.

The model provided in Reviewer Fig. 9 c appears to be arbitrary, and does not exclude a conformational change of the protein. Did the authors test further conformational states?

As indicated in our previous “Answers to the referees”, this model is indeed arbitrary. By no means we are claiming that this conformation reflects the true conformation in presence of GppNHp. Please note that these figures were also not included in the manuscript, but were merely used to show the referees that the particles we are observing in presence of GppNHp do not represent aggregated protein, but can indeed correspond to a monomeric CtRoco protein. Therefore we took two extreme conformations for the monomeric structure: one compact conformation that is just half of the dimer and one elongated particle where the LRR is rotated around the “hinge region”. Potentially a mixture of these and other conformations in the monomeric state are present in solution. The focus of our paper is however not to characterize the detailed conformation of the monomers, but rather to show the dimer-monomer transition.

The immunolabeling experiment in Reviewer Fig. 6 is not designed in a way that allows conclusions. Using anti-rabbit IgG(?) -10 nm gold is not appropriate for such a small particle. Furthermore, the very high background of non-specifically adsorbed antibody can be improved by using appropriate protocols.

A number of techniques to avoid background of non-specifically adsorbed antibodies and gold particles in IEM have been tested as suggested by referee 4 and 5. However, to the best of our knowledge these only apply when looking at cells and tissues or very big particles. A commonly used technique, also suggested by referee 5, is using 0.1% BSA as a blocking agent. When we use this approach for CtRoco, the CtRoco protein can no longer be discerned from the BSA background (BSA: 63 kDa – CtRoco: 127 kDa monomeric; see Reviewer figure 3). Further increasing the washing steps of the carbon-coated copper grid after protein adsorption caused the protein to be washed away completely.

Reviewer Figure 3: Micrograph of 0.01 mg/ml CtRoco after blocking with 0.1% BSA. Nominal magnification: 80000x.

As requested by the referees we performed the gold-labelling experiments using Ni-NTA nanogold directed against the his-tag. Although these experiments confirm the monomer-dimer transition in agreement with all our other experiments, the gold-labelling under conditions compatible with EM is non-stoichiometric and allows only qualitative analysis. We mixed protein and Ni-NTA gold prior to binding to the EM grids. On the grids we observe a distribution of unbound protein, unbound nanogold particles and nanoparticles bound to the proteins. Such a distribution is expected considering: (i) the high multivalency of the gold particles (many Ni-NTA groups per particle) leading to aggregation upon longer incubation times; (ii) the relatively low affinity of Ni-NTA Nanogold for His-tagged proteins combined with the low concentrations; (iii) a certain degree of non-specific binding of the nanogold particles to the grid and to each other.

Nevertheless, after optimization of the protein-gold ratios, the number of washing steps, the incubation time of gold particles and the protein concentration used, we see for nucleotide-free CtRoco that some protein particles clearly have two gold particles bound, a clear indication for dimer formation (Reviewer figure 4a and c). In the presence of GppNHp we only find clear protein particles with maximum one gold particle bound, indicating that only monomeric protein is present (Reviewer figure 4b and d).

We have added the results of these experiments here for the information of the referee. Of course we can add them as a supplementary figure to the manuscript if requested by the referee.

Reviewer Figure 4: Nanogold-labelling of CtRoco. $0.08 \mu\text{M}$ CtRoco is mixed with $0.1 \mu\text{M}$ 5 nm Ni-NTA Nanogold. After 1 min. incubation, $2 \mu\text{l}$ of the sample is spotted on a carbon-coated copper grid, washed and stained. In the GppNHp sample, the protein is incubated for 1h with 1 mM GppNHp prior to the addition of Ni-NTA Nanogold. (a) Micrograph of Nanogold-labelled nucleotide-free CtRoco. The protein particles have two, one or no gold particles bound. (b) Micrograph of Nanogold-labelled GppNHp-bound CtRoco. The particles have one or no Ni-NTA Nanogold bound. (c) Selection of double gold-labelled dimers collected from multiple micrographs. (d) Selection of single gold-labelled monomers collected from multiple micrographs.

Furthermore, it remains unclear whether or not the EM preparation (washing and drying followed by negative staining) could interfere with the conformation and/or nucleotide state.

One can never be absolutely sure whether or not an EM preparation interferes with the protein conformation. Yet it is an established technique and our EM experiments were performed using state-of-the-art staining protocols. Moreover, the class averages give a very good indication that there is no preparation interference, since this would cause random conformations and would not allow for calculating well-defined class averages. Finally, we make a comparison between different nucleotide states which is in very good agreement with observations from our other experiments.

Reviewer #5 (Remarks to the Author):

The manuscript 'A bacterial homologue of the Parkinson's disease-associated protein LRRK2 undergoes a monomer-dimer transition during GTP turnover' from Deyaert et al. reports a novel insight in the activation cycle of the kinase Roc-COR. To understand the function of the monomer-dimer transition and the in vivo significance of the protein is definitively important. The paper is well written. Nevertheless, I'm not convinced by the interpretation of the given results.

The authors try to use a bunch of different methods, SEC-MALS, SEC-SAXS, FRET and EM to understand the conformation and function of CtRoco dependent on his nucleotide state. After changing the manuscript due to given Reviews comments, the results coming from MALS, SAXS and FRET are clearly to understand. The authors are showing a shift of the molecular mass of the protein due to different nucleotide states. The explanation why the dimeric protein with 252kDa elutes at the same volumes as the 173kDa protein (Fig2d) could be explained by a structural change of the monomeric protein. These conformational changes can be postulated on the basis of FRET experiments (Fig 3b).

We thank the referee for this positive comment and we would like to point out that on top of the techniques mentioned by the referee, in the revised version we also used native mass spec, sedimentation velocity AUC and sedimentation equilibrium AUC to convincingly show the dimer/monomer shift upon nucleotide binding.

Nevertheless, the basis of all this evaluations should be a well understood sample. The EM images are due to the addition of nucleotides GTP, GDP and GppNHp of poor quality and I'm not convinced that the images represent the turnover of a dimeric conformation to a functional monomeric protein. The images show clearly a degradation of the dimeric protein after adding nucleotides, shown by an inhomogeneity of the protein shape and sizes in given images. The author concentrates only on structures which reflect possible monomeric and dimeric conformations. All the broken proteins in between are not discussed. There is no proof that the conformation change of the protein due to the addition of nucleotide is specific and do not reflect degradation products of the complex. The turnover experiment (Suppl. Fig 9) indicating a turnover of GTP cycle can as well indicate a destabilization and reorganisation of the complex by the given condition. The author should show that the protein is stable under the given conditions.

We disagree on the first point made by the referee. Apart from EM, we used at least 4 highly analytical techniques to characterize the protein (SAXS, AUC, MALS, MS). All were performed on highly pure sample and show that the protein is well behaved. The EM images in presence of nucleotides are not of poor quality but are state-of-the-art negative stain images for a small particle of only 128 kDa that is in addition flexible (see for example Drake et al (2016) Nature 529:235-8, supplementary Fig. S8, for negative stain EM on a particle of similar size).

Nevertheless, to prove that the protein is stable and does not degrade in the presence of nucleotides, the protein was incubated overnight in the presence of 1 mM GppNHp, 1 mM GDP or without adding nucleotides. After incubation, an SDS-PAGE was run showing no degradation of the protein after the addition of nucleotides (Reviewer figure 5a).

The effect of nucleotides on the thermal stability of CtRoco was further confirmed using a Thermal Shift Assay either with or without nucleotides. As can be seen in Reviewer figure 5b, the melting temperatures are very similar for nucleotide-free and GDP-bound protein and even increases in the presence of GppNHp. As such the protein clearly does not unfold upon addition of nucleotides. From time-resolved EM imaging it can be seen that monomerization is a reversible process (Figure 4c), because dimers reappear after GTP is hydrolysed. This behaviour is also incompatible with degradation.

Finally, Reviewer figure 5c shows a TEM micrograph of the protein after the addition of 8M urea. This proves that EM images of denatured CtRoco clearly differ from what is observed after nucleotide addition.

The referee also suggests that we only focussed on certain particles for making class averages. We would like to clarify that the class averages shown in figure 2 are the result of particles picked from micrographs generated in the absence or presence of 1 mM nucleotide. In these micrographs all single particles were selected in an unbiased way. A possible misunderstanding could be caused by the boxed particles on the micrographs of Figure 2f. However, these boxed particles just represent a few representative particles that we show magnified underneath the micrograph in Figure 2f. However, for making class averages all particles were taken in consideration without any "cherry picking". If requested, we are happy to share our raw micrographs with the referee.

Reviewer figure 5: Stability of CtRoco full length protein. (a) SDS-PAGE of CtRoco after overnight incubation with 1 mM nucleotides or in absence of nucleotides (NF: nucleotide-free). (b) Thermal shift assay performed on CtRoco in the absence or presence of 1 mM nucleotides. The temperature was gradually increased. After fitting on a Boltzmann sigmoidal curve, the melting temperature was determined. Each data point is an average of 3 independent measurements. (c) Transmission Electron Microscopy micrograph of nucleotide-free CtRoco after incubation in 8M urea. The protein is completely denatured. Nominal magnification: 80 000x.

The class averages of the particles given in Fig.2 are not well defined and centred. They reflect sometimes a combination of more than one protein. I do not agree with the author validations, that the reported EM images in the manuscript reflect a conformational change, proofed by a comparison of the class averages with theoretical 2D EM projections given in Reviewer Fig. 1c. The theoretical EM images do not correspond

very well to the experimentally observed class averages of the free dimeric CtRoco, even under the argument of missing 150 C-terminal amino acids.

We do not agree with the referee that the class averages are not well defined and centred. As a response to the referee's concern that the class averages are a combination of more than one protein, we show in Reviewer figure 6 the first class average of CtRoco GppNHp and the raw particles constituting the class, as an example. In this way it can be ascertained that the extra density attached to the 'wormlike' particle in the class average can also be observed in the single particles and is not due to neighbouring particles.

For the remark regarding the differences observed between the theoretical projections and the experimentally observed class averages, we would like to refer to our answer to reviewer 4. We would just like to reiterate that from SAXS, MALS, MS, AUC and FRET it is clear that the protein undergoes a dimer/monomer transition. The EM images clearly support this. A comparison between the theoretical projections and class averages was only used to show the referees that the particles we see can account for a monomeric and dimeric protein. This comparison is also not included in the manuscript. Obviously, there can be no perfect fit since the structure of the full length CtRoco protein is not known.

Reviewer figure 6: Example of a CtRoco GppNHp class average and the single particles constituting the class. (a) First class average of CtRoco GppNHp. (b) The single particles constituting the class. For many particles small densities disconnected from the 'worm-shaped' 'body' of the particles can be observed, which are clearly not a part of the neighbouring particle.

The superposition of the crystallographic model to the given ab initio SAXS model is still not very informative.

The envelope of the CtRocCOR shows that there is a very good agreement with the crystal structure (considering that SAXS is a low resolution method). The envelope of the full length protein shows the extra features corresponding to the additional domains. We added the latter envelope on request of another referee in the previous revision round.

To proof the nature of the protein conformation IEM should be performed. The argument, that the Immunogold labelling causes an unspecific background is not valid. There are enough possibilities to deal with this background and get rid of it (Blocking reagents like BSA, wash steps, usage of Protein A gold, higher dilutions).

Since this was also a remark of reviewer 4, we repeat below our answer.

A number of techniques to avoid background of non-specifically absorbed antibodies and gold particles in IEM have been tested as suggested by referee 4 and 5. However, to the best of our knowledge these only apply when looking at cells and tissues or very big particles. A commonly used technique, also suggested by referee 5, is using 0.1% BSA as a blocking agent. When we use this approach for CtRoco, the CtRoco protein can no longer be discerned from the BSA background (BSA: 63 kDa – CtRoco: 127 kDa monomeric; see Reviewer figure 3). Further increasing the washing steps of the carbon-coated copper grid after protein adsorption caused the protein to be washed away completely.

Reviewer Figure 3: Micrograph of 0.01 mg/ml CtRoco after blocking with 0.1% BSA. Nominal magnification: 80000x.

As requested by the referees we performed the gold-labelling experiments using Ni-NTA nanogold directed against the his-tag. Although these experiments confirm the monomer-dimer transition in agreement with all our other experiments, the gold-labelling under conditions compatible with EM is non-stoichiometric and allows only qualitative analysis. We mixed protein and Ni-NTA gold prior to binding to

the EM grids. On the grids we observe a distribution of unbound protein, unbound nanogold particles and nanoparticles bound to the proteins. Such a distribution is expected considering: (i) the high multivalency of the gold particles (many Ni-NTA groups per particle) leading to aggregation upon longer incubation times; (ii) the relatively low affinity of Ni-NTA Nanogold for His-tagged proteins combined with the low concentrations; (iii) a certain degree of non-specific binding of the nanogold particles to the grid and to each other.

Nevertheless, after optimization of the protein-gold ratios, the number of washing steps, the incubation time of gold particles and the protein concentration used, we see for nucleotide-free CtRoco that some protein particles clearly have two gold particles bound, a clear indication for dimer formation (Reviewer figure 4a and c). In the presence of GppNHp we only find clear protein particles with maximum one gold particle bound, indicating that only monomeric protein is present (Reviewer figure 4b and d).

We have added the results of these experiments here for the information of the referee. Of course we can add them as a supplementary figure to the manuscript if requested by the referee.

Reviewer Figure 4: Nanogold-labelling of CtRoco. $0.08 \mu\text{M}$ CtRoco is mixed with $0.1 \mu\text{M}$ 5 nm Ni-NTA Nanogold. After 1 min. incubation, $2 \mu\text{l}$ of the sample is spotted on a carbon-coated copper grid, washed and stained. In the GppNHp sample, the protein is incubated for 1h with 1 mM GppNHp prior to the addition of Ni-NTA Nanogold. (a) Micrograph of Nanogold-labelled nucleotide-free CtRoco. The protein particles have two, one or no gold particles bound. (b) Micrograph of Nanogold-labelled GppNHp-bound CtRoco. The particles have one or no Ni-NTA Nanogold bound. (c) Selection of double gold-labelled dimers collected from multiple micrographs. (d) Selection of single gold-labelled monomers collected from multiple micrographs.

I fully agree with the author, that the question if CtRoco undergoes dimer-monomer cycle is important, but I do not agree that the data clearly indicates that the monomeric state of LRRK2 exists and leads to a new model of monomer-dimer transition. The manuscript therefore is acceptable but needs additional work. I

would consider that add additional proof of the stability of the sample before and after adding nucleotide. Immunogold labelling should be performed. The interpretation of EM data should be done with purified and stabilized samples. In vivo data showing the abundance of monomeric-dimeric conformations, as well as the location of these proteins should be added.

See our answers above to address the remarks concerning stability and nanogold labelling.

Since this paper focuses entirely on the detailed in vitro characterization of CtRoco we would like to argue that an in vivo study as suggested by the referee falls outside the scope of this paper.

REVIEWERS' COMMENTS:

Reviewer #5 (Remarks to the Author):

Manuscript No.: NCOMMS-16-22070A

A bacterial homologue of the Parkinson's disease-associated protein LRRK2 undergoes a monomer-dimer transition during GTP turnover

Point-by-point response to referee remarks

The authors fully addressed to my concerns of the manuscript. They provided additional information and results to clarify my major criticism, that the monomer/dimer cycle could be degradation of protein. Therefore I'm fully satisfied with their reply.

Due to the response of the authors I agree with a publication of the study in Nature Communications.